# RADAR: FAST LONG-CONTEXT DECODING FOR ANY TRANSFORMER

**Yongchang Hao**[*]
University of Alberta & RBC Borealis
`yongcha1@ualberta.ca`

**Mengyao Zhai**
RBC Borealis
`mengyao.zhai@borealisai.com`

**Hossein Hajimirsadeghi**
RBC Borealis
`hossein.hajimirsadeghi@borealisai.com`

**Sepidehsadat Hosseini**
RBC Borealis
`sepid.hosseini@borealisai.com`

**Frederick Tung**
RBC Borealis
`frederick.tung@borealisai.com`

## ABSTRACT

Transformer models have demonstrated exceptional performance across a wide range of applications. Though forming the foundation of Transformer models, the dot-product attention does not scale well to long-context data since its time requirement grows quadratically with context length. In this work, we propose Radar, a training-free approach that accelerates inference by dynamically searching for the most important context tokens. For any pre-trained Transformer, Radar can reduce the decoding time complexity without training or heuristically evicting tokens. Moreover, we provide theoretical justification for our approach, demonstrating that Radar can reliably identify the most important tokens with high probability. We conduct extensive comparisons with the previous methods on a wide range of tasks. The results demonstrate that Radar achieves the state-of-the-art performance across different architectures with reduced time complexity, offering a practical solution for efficient long-context processing of Transformers. The code is publicly available at `https://github.com/BorealisAI/radar-decoding`.

## 1 INTRODUCTION

Transformer models demonstrate an extraordinary ability on different sequential processing tasks, including language modeling (Vaswani et al., 2017; Devlin et al., 2019; Raffel et al., 2020; Touvron et al., 2023b), image classification (Dosovitskiy et al., 2021; Liu et al., 2021; Hassani et al., 2022; Zhu et al., 2023), translation (Tang et al., 2020; Yao & Wan, 2020), and many more (Carion et al., 2020; Ding et al., 2022; Chang et al., 2023; Xu et al., 2022; Fang et al., 2023). In particular, Transformer models take each input as a sequence of tokens and compute the embedding of each token for downstream tasks. Among all components, the dot-product attention has been shown to be critical to the success of Transformer models (Choromanski et al., 2021). It not only enables parallel computation of sequences during training (Vyas et al., 2020), but also provides a high-quality method for sequence modeling (Sanford et al., 2023).

Despite being at the core of Transformer models, the dot-product attention is not ideal for long-context data: the time to process each token increases with context lengths, significantly slowing down the throughput on long-context data. Moreover, the maximum context length is limited during training, resulting in an inability to perform inference on long-context tasks. Yet, many real-world applications are naturally long-context (Tay et al., 2021; Beltagy et al., 2020; Wu et al., 2024). For example, a code file could have more than 10K tokens (Lozhkov et al., 2024; Kocetkov et al., 2022).

---

[*]Work done during an internship at RBC Borealis.

However, Llama-2 (Touvron et al., 2023b), a widely used Transformer-based model, is incapable of processing the full source code because it has a maximum context length of 4K tokens.

To improve Transformer's long-context performance, a line of research proposes to replace the original attention with some faster variants (Xiao et al., 2024; Peng et al., 2021; Bertsch et al., 2023; Beltagy et al., 2020; Choromanski et al., 2021; Mohtashami & Jaggi, 2023; Peng et al., 2023). One notable example is attention kernelization, which reformulate the dot-product attention as a kernel and approximate the kernel with linear operations (Peng et al., 2021). By reordering the matrix chain multiplication, kernelization methods reduce the time complexity to $O(t)$ for the context of length $t$. However, such methods require a training process to fit the approximations, which becomes more and more infeasible given the fast parameter scaling of the current Transformer models.

Without the need for training, another line of work accelerates inference by evicting previous tokens used in each decoding step (Xiao et al., 2024; Zhang et al., 2023; Li et al., 2024). For instance, StreamingLLM (Xiao et al., 2024) proposes to only use the constant-length recent tokens and discard all other but the first tokens. Although they are faster in inference on long-context data, these methods lose previous information in the context that is potentially critical for future use. Using the code file as an example, StreamingLLM could fail to invoke a function if the function declaration is not in the recent tokens.

In this paper, we propose Radar (**r**ange se**a**rch accelerate**d** by **r**andom featu**r**es), a training-free approach that dynamically selects the important context ranges to accelerate inference. Specifically, Radar groups the context into segments and maintains a "summarization representation" for each group by random projections. To generate a new token, Radar calculates the importance of each group and selects only the tokens in the most important segments. We theoretically justify our approach by showing that Radar is guaranteed to pick the most dominant tokens with high probabilities. For any pre-trained Transformers, our approach can instantly extend the maximum context length while reducing the overall complexity to $O(t^{1.5})$.

We conduct extensive experiments to verify the effectiveness of our approach. Specifically, we compared our method with StreamingLLM (Xiao et al., 2024), Landmark attention (Mohtashami & Jaggi, 2023), H$_2$O (Zhang et al., 2023), and SnapKV (Li et al., 2024). The results across different models and tasks show that Radar generally achieves state-of-the-art performance with a low time complexity. Additionally, our in-depth analyses, while providing detailed information, highly corroborate our main results.

## 2 METHODOLOGY

In this section, we present Radar, a principled approach that is guaranteed to select the most important tokens with a high probability. We first introduce the notations and the problem formulation in Section 2.1. The algorithm of Radar is formally presented in Section 2.2 with a major focus on showing the time complexity. The correctness of our algorithm is rigorously shown in Section 2.3.

### 2.1 BACKGROUND

**Dot-product attention.** Without loss of generality, we focus on the single-head self-attention for simplicity. Specifically, it operates on the level of sequence. Each sequence contains several token embeddings $(\boldsymbol{x}_1, \ldots, \boldsymbol{x}_t)$, where every embedding is a vector in the high dimensional space $\mathbb{R}^d$. The attention first maps each embedding to query, key, and values by linear projections ($\boldsymbol{q}_i = \boldsymbol{W}_q \boldsymbol{x}_i$, $\boldsymbol{k}_i = \boldsymbol{W}_k \boldsymbol{x}_i$, and $\boldsymbol{v}_i = \boldsymbol{W}_v \boldsymbol{x}_i$). The attention score is defined as

$$a_{i,j} := \frac{1}{z_i}\mathbb{I}(j \leq i)\exp\left(\boldsymbol{q}_i^\top \boldsymbol{k}_j / \sqrt{d}\right) \tag{1}$$

for $1 \leq i, j \leq t$. Here, $z_i$ is a normalization factor such that $\sum_{j=1}^{i} a_{i,j} = 1$. The dot-product attention then produces a sequence by

$$\boldsymbol{o}_i := \sum_{j=1}^{i} a_{i,j}\boldsymbol{v}_j, \tag{2}$$

indicating that $\boldsymbol{o}_i$ is a fusion of $(\boldsymbol{v}_1, \ldots, \boldsymbol{v}_i)$ weighted by the attention scores $(a_{i,1}, \ldots, a_{i,i})$.

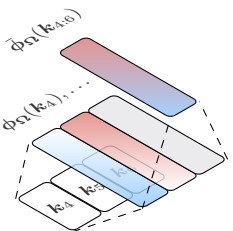
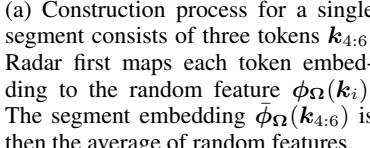
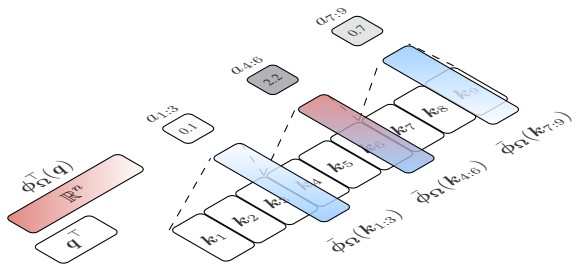

(a) Construction process for a single segment consists of three tokens $\boldsymbol{k}_{4:6}$. Radar first maps each token embedding to the random feature $\boldsymbol{\phi}_{\boldsymbol{\Omega}}(\boldsymbol{k}_i)$. The segment embedding $\bar{\boldsymbol{\phi}}_{\boldsymbol{\Omega}}(\boldsymbol{k}_{4:6})$ is then the average of random features.

(b) Query process for a single query $\boldsymbol{q}$. The query token is first mapped to the random feature $\boldsymbol{\phi}_{\boldsymbol{\Omega}}(\boldsymbol{q})$. The segment importance is obtained by inner products between random features $\boldsymbol{\phi}_{\boldsymbol{\Omega}}^{\top}(\boldsymbol{q})\bar{\boldsymbol{\phi}}_{\boldsymbol{\Omega}}(\boldsymbol{k}_{i:i+2})$. In this example, the second segment has the highest (unnormalized) segment attention of 2.2.

Figure 1: Overview of the approach.

The two equations above show that the dot-product attention requires $O(t)$ to generate a new token, resulting in an overall $O(t^2)$ time for the whole sequence. To reduce the complexity of the dot-product attention, both operations should not be quadratic in $t$.

**Context reduction.** Without the need for training, one popular strategy for accelerating inference is to reduce the number of tokens used in attention calculation. LongFormer (Beltagy et al., 2020) proposes to use local sliding window attention, which only needs to look into the tailing tokens, to accelerate decoding. More recently, StreamingLLM (Xiao et al., 2024) improves this by prepending sink tokens to stabilize the decoding for contexts beyond the pre-training length. However, these methods suffer from the issue of *information loss*, where the vast amount of middle tokens potentially containing important information are permanently lost. Other remedies such as the heavy-hitter oracle ($H_2O$) use heuristics to partially retain some middle tokens. Despite this, they depend on heuristic rules and sometimes fail to completely prevent information loss.

**Problem formulation.** Given Equation (2), the attention score $a_{i,j}$ represents the importance of the $j$th token to the current step $i$. To ensure the selected tokens are important, we would like to select an index set $\mathbb{S}$ that

$$j \in \underset{1 \le l \le i}{\arg\operatorname{topk}} \, a_{i,l} \tag{3}$$

for all $j \in \mathbb{S}$. This gives us an approximation of the original attention and reduces the compute to $O(|\mathbb{S}|)$. However, generating such an index set takes $O(t)$ time, failing to improve the overall time complexity. We aim to approximately solve this problem in sublinear time complexity.

## 2.2 THE PROPOSED ALGORITHM: RADAR

This section shows the algorithm of Radar with a focus on analyzing the time complexity. The correctness of the algorithm is deferred to Section 2.3.

**Hierarchical structure.** We manage the key embeddings $(\boldsymbol{k}_1, \ldots, \boldsymbol{k}_t)$ with a hierarchical data structure consisting of two layers of nodes. The bottom-layer nodes are the original key embeddings, whereas each top-layer node is a *segment* consisting of the bottom keys. The hierarchical structure enables a two-step attention calculation procedure: for the first step, we search for the important segments with the highest summed attention score; after that, we use the tokens in the selected segments to approximate the attention for attention calculation.

The benefit of applying the hierarchical paradigm is reducing the time complexity. Assume there are $t$ tokens and each segment contains $c$ tokens on average, then the overall query complexity is $O(\tau(c) \cdot t/c + c)$, where $O(\tau(c))$ is the time complexity to obtain the importance score for a segment. The term $O(t/c)$ comes from calculating the importance of $\lceil t/c \rceil$ segments and selecting the top ones; and the second term $O(c)$ is for the actual attention by Equations (1) and (2).

**Accelerated segment search.** To accelerate the segment search, we propose to assign each segment node a representation that "summarizes" the bottom-layer key embeddings. We resort to the attention kernelization technique to accomplish this. Specifically, we first apply a mapping $\phi_{\boldsymbol{\Omega}} : \mathbb{R}^d \to \mathbb{R}^n$ in (Choromanski et al., 2021) for all keys

$$\phi_{\boldsymbol{\Omega}}(\boldsymbol{k}) := \frac{1}{\sqrt{n}}\Big( \exp\big(\boldsymbol{\omega}_1^\top \boldsymbol{k}' - \|\boldsymbol{k}'\|_2^2/2\big), \dots, \exp\big(\boldsymbol{\omega}_n^\top \boldsymbol{k}' - \|\boldsymbol{k}'\|_2^2/2\big)\Big). \tag{4}$$

Here, every element in $\boldsymbol{\Omega} \in \mathbb{R}^{n \times d}$ is sampled from $\mathcal{N}_{\boldsymbol{0},\boldsymbol{1}}$ and the vector $\boldsymbol{\omega}_i \in \mathbb{R}^d$ is the $i$th column of $\boldsymbol{\Omega}$. We let $\boldsymbol{k}' := \boldsymbol{k}/\sqrt[4]{d}$ for the scaling in attention. After projection, we average the projected features as the segment embedding. The embedding for a segment ranging from $i$ to $i + c$ is

$$\bar{\phi}_{\boldsymbol{\Omega}}(\boldsymbol{k}_{i:i+c}) := \frac{1}{c}\sum_{l=0}^{c-1}\phi_{\boldsymbol{\Omega}}(\boldsymbol{k}_{i+l}). \tag{5}$$

The segment embeddings can be pre-computed and stored. This construction process is illustrated in Figure 1a.

To search for the most important segments, we can search for

$$\arg\operatorname*{topk}_{l}\big\{a_{l:l+c} := \phi_{\boldsymbol{\Omega}}^\top(\boldsymbol{q})\bar{\phi}_{\boldsymbol{\Omega}}(\boldsymbol{k}_{l:l+c})\big\} \tag{6}$$

for $l \in \{1, 1+c, \dots, 1 + (\lceil t/c \rceil - 1)c\}$. By using this acceleration, the time for calculating the importance score of a single segment (i.e., $\tau(c)$) becomes constant, and consequently $O(\tau(c)) = O(1)$. Therefore, the total time is $O(t/c)$ given there are $\lceil t/c \rceil$ segments. Based on our theoretical guarantees in Section 2.3, the obtained segments will likely contain the most important tokens that summed to the highest attention scores. This process is demonstrated in Figure 1b.

**Dynamic restructuring.** With the accelerated searching, we have $O(t/c + c)$ time for each query. The optimal asymptotic time complexity $O(\sqrt{t})$ is obtained when $c = O(\sqrt{t})$, indicating that the segment range should change with the length of the context. However, changing the segment range requires a reconstruction of the segment embeddings that takes $O(t)$ time. To address this, we propose a dynamic restructuring schedule, which only performs reconstruction when $\sqrt{t} \in \mathbb{N}$. This schedule indicates that there will be maximum $\sqrt{t}$ times of restructuring happening for $t$ tokens, amortized to an $O(\sqrt{t})$ time for each query step.

For all other steps that $\sqrt{t} \notin \mathbb{N}$, we store the token in a buffer $\mathbb{W}$, which serves as a sliding window with a maximum size of $2\sqrt{t}$ - 1. The sliding window is always used in querying steps. Although this adds more tokens in the attention calculation, the asymptotic complexity remains $O(\sqrt{t})$ for each step.

**The overall algorithm.** Putting all together, we see that Radar demonstrates an overall time complexity of $O(t^{1.5})$ to generate $t$ tokens. We provide the procedure of Radar with detailed time complexity analysis in Appendix A.

## 2.3 THEORETICAL JUSTIFICATIONS

In this section, we rigorously show that our algorithm described in Section 2.2 is guaranteed to solve the problem in Equation (3) with high probability.

We first show that the projection 4 is sound with the following lemma.

**Lemma 1.** *Let $\phi_{\boldsymbol{\Omega}}$ follow the definition* (4) *where $\boldsymbol{\Omega} = (\boldsymbol{\omega}_1, \dots, \boldsymbol{\omega}_n)$ is sampled from $\mathcal{N}_{\boldsymbol{0},\boldsymbol{1}}$. Given any $\boldsymbol{u}, \boldsymbol{v} \in \mathbb{R}^d$, we have $\mathbb{E}_{\boldsymbol{\Omega}}[\phi_{\boldsymbol{\Omega}}^\top(\boldsymbol{u})\phi_{\boldsymbol{\Omega}}(\boldsymbol{v})] = \exp\Big(\boldsymbol{u}^\top \boldsymbol{v}/\sqrt{d}\Big)$.*

*Proof.* This is first shown in (Choromanski et al., 2021). We include as a part of our Lemma 6 in Appendix B for completeness. $\square$

Based on this, we build the following theorem to show the correctness of our algorithm.

**Theorem 2.** *Let Radar be at a state of $c$ segments where each segment contains $c$ tokens. Without loss of generality, assume the first segment $\boldsymbol{k}_{1:1+c}$ has the highest attention score $a_{1:1+c}$ w.r.t. the current query $\boldsymbol{q}$. With confidence at least $1 - \delta$, we have*

$$\phi_{\boldsymbol{\Omega}}^{\top}(\boldsymbol{q})\bar{\phi}_{\boldsymbol{\Omega}}(\boldsymbol{k}_{1:c+1}) = \max_l \phi_{\boldsymbol{\Omega}}^{\top}(\boldsymbol{q})\bar{\phi}_{\boldsymbol{\Omega}}(\boldsymbol{k}_{l:c+l}) \tag{7}$$

*if the minimum gap $a_{1:1+c} - \max_l a_{l:l+c} \geq \frac{1}{c}\exp\left(\zeta^2/\sqrt{d}\right)\sqrt{\frac{8\log(2(c-1)/\delta)}{n}}$. Here, $\zeta$ is the maximum norm among $\boldsymbol{k}_i$ and $\boldsymbol{q}$, whereas $l \in [1, 1+c, 1+2c, \ldots, 1+(c-1)c]$ is the starting index of each segment.*

*Proof.* We first build a series of essential lemmas in Appendix B. The proof of this theorem is then shown in Appendix C. $\qquad\square$

Our theorem asserts that as long as the projection dimension $n$ is large enough, we can correctly retrieve the segment with the highest attention up to any precision. In fact, with the growth of the sequence length, the difficulty of meeting this condition is not growing linearly because the attention gap needed is $O(\log(c)/c)$ for an average segment attention score of $O(1/c)$, suggesting our approach is particularly favorable for long-context decoding.

## 3 EXPERIMENTS

In this section, we conduct extensive experiments to verify the effectiveness of Radar in comparison with other methods on diverse tasks and models. We additionally carry out in-depth analyses and ablation studies to demonstrate the effects of the length of prompts, the projection dimension, and the number of selected segments.

### 3.1 EVALUATING LONG-CONTEXT PERPLEXITY

**Datasets.** Following previous work (Xiao et al., 2024; Zhang et al., 2023), we use the perplexity on the first sample in the PG-19 dataset as the main test bed. Specifically, we evaluate the overall perplexity by feeding the ground-truth tokens one by one. Since PG-19 only contains natural language, we additionally use a code sample from The Stack (Lozhkov et al., 2024) for a broader comparison. The selected sample is an implementation of NASA's CDF Project[1] in a single Python file. To simulate the real-world use cases, we prefill the first 16,384 tokens into the model as a prompt. The experiments without prompts are conducted in Section 3.3.

**Competing methods.** We consider the vanilla dot-product attention (Vaswani et al., 2017) and StreamingLLM (Xiao et al., 2024) as the baselines in this experiment. Following the default setting in previous work (Xiao et al., 2024; Zhang et al., 2023) the sliding window length is set to 1024 for all runs. For our Radar, we choose the top 64 segments for each query with the random projection dimension set to 2048. The effect of these two parameters is studied in Section 3.3. Each experiment is conducted on a single A100 GPU with 40GB of memory.

**Models.** Following previous work (Xiao et al., 2024), we choose two popular architectures in this experiment: Llama (Dubey et al., 2024) and Mistral (Jiang et al., 2023). Specifically, we use `Llama-Meta-3.1-8B` and `Mistral-7B-v0.3` for each architecture, respectively.

**Results.** We show the results in Figure 2. The naive attention method achieves the best perplexity in all runs. However, its elapsed time grows quadratically regardless of the architecture. In practice, the generation throughput drops to 10 tokens/s or even lower at the end. On the other hand, StreamingLLM achieves a constantly high throughput at a cost of poor generation quality (measured in perplexity), as it evicts most of the context during generation. Our method Radar, in contrast, is able to effectively control the elapsed time while maintaining a similar perplexity by a maximum

---

[1]`https://cdf.gsfc.nasa.gov/`

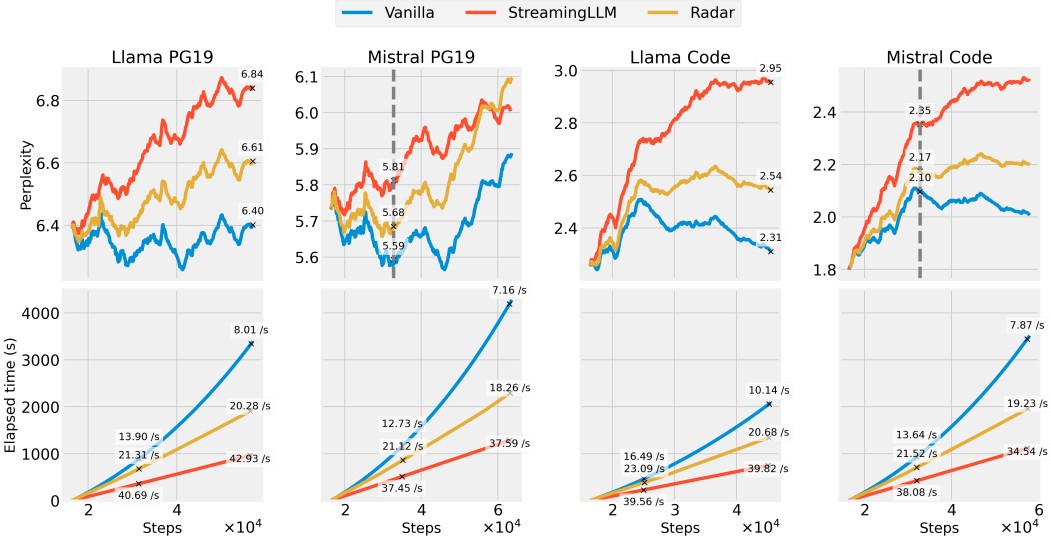

Figure 2: The performance comparison in perplexity (first row) and elapsed time (second row). The lower the better for both metrics. For the Llama model, we annotate the perplexity value at the last token; for the Mistral model, we annotate the perplexity at the maximum pre-training context length (shown by the vertical dashed lines) because the full context exceeds its modeling ability. We additionally show the generation throughput for all runs.

difference of around 0.2 compared with the vanilla attention. Notably, the performance degeneration of Radar is at most 10% while the speed up is more than 2 times. The results confirm the effectiveness of our method in effectively utilizing the context while accelerating inference.

We present additional results with $H_2O$ (Zhang et al., 2023) and SnapKV (Li et al., 2024), two state-of-the-art methods that employ different heuristics to reduce context length during inference, in a separate figure in Appendix D as both methods produce out-of-scale perplexities in this setting.

## 3.2 LONG-CONTEXT BENCHMARKS

In Section 3.1 above, we only evaluated the perplexity without inspecting the down-stream performance. For a more comprehensive comparison, we additionally conduct the end-to-end evaluation on numerous long-context tasks.

**Datasets.** Following previous work (Li et al., 2024), we use the LongBench dataset (Bai et al., 2024) as the main benchmark. Specifically, we use all 16 subtasks in English and code languages. These tasks belong to 6 categories in total: single-doc QA, multi-doc QA, summarization, few-shot learning, synthetic, and code. Following the standard LongBench pipeline (Bai et al., 2024), we truncate the context from the middle of the prompt length exceeds the pre-training length. Each subtask is evaluated with a specific metric (e.g., ROUGE scores (Lin, 2004) for summarization tasks and accuracies for synthetic tasks). After evaluating all subtasks, we report the average scores as a summary on LongBench. It is worth noting that the metrics used by different subtasks can have very different scales, indicating the arithmetic average over all subtasks may not reflect the overall performance (e.g., the final average may be inaccurately dominated by the accuracy of the synthetic tasks as pointed out by the original paper (Bai et al., 2024)). To this end, we additionally include the average percentile over all 16 tasks ranked within the same base model. For instance, a percentile of 80% means that the method is expected to be strictly better than 80% of other methods.

**Competing methods.** We use all methods mentioned in Section 3.1, including $H_2O$ (Zhang et al., 2023) and SnapKV (Li et al., 2024). Following the previous work (Li et al., 2024), all competing baselines can use a maximum of $32 + n_c$ tokens, where 32 is the length of the sliding window and $n_c$ is the number of middle tokens varied from 1024 to 4096. For StreamingLLM, we simply extend the sliding window by $n_c$. In addition to baselines used in previous work, we also evaluate

Table 1: Performance comparisons of different methods on LongBench. On each model, the best-performing method is highlighted in **bold** and the second-best method is underlined (excluding the vanilla method). Each table contains the results for one particular model.

(a) The LongBench benchmark results of different methods applied on `Llama-7b`. Each prompt contains maximum 1.5K tokens as the model can handle maximum 2K tokens.

| Context | Method | Single QA | | | Multi QA | | | Summarization | | | Few-show | | | Synthetic | | Code | | Avg. Score | Avg. Perc. |
|---|---|---|---|---|---|---|---|---|---|---|---|---|---|---|---|---|---|---|---|
| | | NrtvQA | Qasper | MFQA | HtptQA | 2WkQA | Musique | GovRep | QMSum | MulNews | TREC | TrivQA | SamSum | PsgCnt | PsgRet | LCC | RB-P | | |
| Full | Vanilla | 3.12 | 8.01 | 17.52 | 8.16 | 10.02 | 4.75 | 26.28 | 14.33 | 27.18 | 52.50 | 80.88 | 35.16 | 2.00 | 6.17 | 61.88 | 54.81 | 25.80 | 56.25 |
| - | Landmark | **8.53** | **10.51** | 16.05 | **10.97** | **12.61** | 4.54 | 11.93 | 11.88 | 7.16 | 34.00 | 58.83 | 28.93 | **2.50** | 5.83 | 45.24 | 39.39 | 19.31 | 30.21 |
| 1024 | StreamingLLM | 2.76 | 7.86 | 16.37 | 8.57 | 11.14 | 4.54 | 15.86 | 14.17 | 19.96 | **53.00** | 80.79 | 35.44 | 1.83 | **6.50** | 61.81 | 53.97 | 24.66 | 37.50 |
| | H$_2$O | 2.87 | 8.11 | 16.59 | 8.05 | 9.85 | 4.20 | **26.25** | **14.85** | 25.92 | 51.50 | 80.67 | 34.41 | 1.50 | 5.75 | 60.06 | 52.96 | 25.22 | 25.00 |
| | SnapKV | 2.97 | 7.88 | 17.38 | 8.39 | 10.07 | **4.65** | 25.74 | 14.51 | 26.55 | 52.50 | 80.88 | 35.24 | 2.00 | 6.17 | **62.71** | **54.47** | 25.76 | 52.08 |
| | Radar | 2.94 | 8.01 | **17.61** | 8.49 | 10.51 | 4.15 | 26.11 | 14.61 | **27.52** | 52.00 | **80.92** | **35.48** | 1.75 | **6.50** | 62.36 | 54.41 | **25.84** | **54.17** |

(b) The LongBench benchmark results of different methods applied on `Llama-2-7b-chat-hf`. Each prompt contains maximum 3.5K tokens as the model can handle maximum 4K tokens.

| Context | Method | Single QA | | | Multi QA | | | Summarization | | | Few-shot | | | Synthetic | | Code | | Avg. Score | Avg. Perc. |
|---|---|---|---|---|---|---|---|---|---|---|---|---|---|---|---|---|---|---|---|
| | | NrtvQA | Qasper | MFQA | HtptQA | 2WkQA | Musique | GovRep | QMSum | MulNews | TREC | TrivQA | SamSum | PsgCnt | PsgRet | LCC | RB-P | | |
| Full | Vanilla | 16.87 | 17.78 | 36.13 | 34.06 | 27.53 | 9.10 | 26.09 | 21.01 | 25.99 | 64.00 | 83.84 | 41.24 | 4.50 | 12.00 | 58.36 | 52.31 | 33.18 | 74.38 |
| 1024 | SubGen | 15.98 | **19.54** | 26.27 | 17.00 | 21.49 | 7.38 | 22.80 | 20.68 | 23.90 | 39.00 | 65.31 | 25.95 | 1.84 | 4.92 | 44.23 | 43.32 | 24.98 | 13.75 |
| | StreamingLLM | 15.38 | 15.13 | 21.95 | 28.67 | 24.56 | 5.66 | 21.15 | 19.70 | 24.57 | 61.00 | 80.69 | 40.62 | **4.58** | 4.50 | 56.59 | 49.28 | 29.63 | 14.37 |
| | H$_2$O | 16.54 | 16.04 | 35.14 | 32.53 | 26.60 | 8.11 | **27.91** | 20.42 | **26.84** | 63.00 | 83.15 | 38.38 | 4.00 | 10.50 | 52.11 | 44.53 | 31.61 | 36.25 |
| | SnapKV | 15.69 | 18.30 | 35.59 | 33.23 | 26.27 | 8.62 | 21.81 | 20.47 | 25.18 | **64.50** | 82.57 | 40.43 | 4.50 | 11.00 | 58.27 | 52.06 | 32.41 | 45.00 |
| | Radar | 16.95 | 19.32 | **37.20** | 33.70 | **27.60** | **8.83** | 25.30 | **21.21** | 25.66 | 63.50 | 83.44 | 40.78 | 4.50 | 10.00 | 57.70 | 51.17 | 32.93 | 64.38 |
| 2048 | StreamingLLM | 16.77 | 16.03 | 25.45 | 30.14 | 25.36 | 7.42 | 23.59 | 19.71 | 25.58 | 64.00 | 82.57 | **41.65** | 4.50 | 6.50 | 56.94 | 51.15 | 31.08 | 35.00 |
| | H$_2$O | 16.39 | 17.93 | 36.09 | 32.88 | 26.47 | 7.61 | 27.73 | 20.58 | 26.45 | 63.50 | 83.78 | 39.26 | 4.50 | 10.00 | 57.32 | 51.63 | 32.63 | 50.00 |
| | SnapKV | **17.05** | 18.43 | 36.03 | 33.78 | 27.06 | 7.90 | 24.56 | 21.01 | 25.76 | 64.00 | 82.88 | 40.60 | 4.50 | **11.50** | **58.51** | 51.74 | 32.83 | 64.38 |
| | Radar | 16.90 | 17.76 | 36.02 | **33.90** | 26.81 | 8.79 | 26.32 | 21.11 | 26.15 | 64.00 | **83.92** | 41.01 | 4.50 | **11.50** | 58.37 | **52.06** | **33.07** | **71.25** |

(c) The LongBench benchmark results of different methods applied on `Mistral-7B-Instruct-v0.2`. Each prompt contains maximum 31.5K tokens as the model can handle maximum 32K tokens.

| Context | Method | Single QA | | | Multi QA | | | Summarization | | | Few-shot | | | Synthetic | | Code | | Avg. Score | Avg. Perc. |
|---|---|---|---|---|---|---|---|---|---|---|---|---|---|---|---|---|---|---|---|
| | | NrtvQA | Qasper | MFQA | HtptQA | 2WkQA | Musique | GovRep | QMSum | MulNews | TREC | TrivQA | SamSum | PsgCnt | PsgRet | LCC | RB-P | | |
| Full | Vanilla | 27.06 | 32.22 | 49.61 | 43.49 | 27.96 | 18.85 | 33.35 | 24.30 | 27.13 | 71.00 | 86.07 | 43.50 | 2.80 | 86.98 | 56.17 | 53.63 | 42.76 | 75.63 |
| 1024 | StreamingLLM | 21.70 | 18.95 | 32.03 | 32.72 | 21.69 | 11.89 | 23.55 | 20.28 | 25.41 | 64.00 | 84.95 | 42.13 | 3.13 | 22.33 | 54.79 | 49.94 | 33.09 | 8.75 |
| | SnapKV | 24.97 | 30.03 | 49.51 | 41.35 | 25.82 | **18.92** | 25.85 | 24.09 | 26.24 | 70.00 | **86.53** | 42.04 | 2.83 | **89.06** | 54.33 | 50.96 | 41.41 | 43.75 |
| | Radar | 23.81 | 28.26 | 47.91 | 38.96 | 26.17 | 16.08 | 27.99 | 22.52 | 26.95 | 70.50 | 85.94 | 41.70 | 3.13 | 53.33 | 54.17 | 50.03 | 38.59 | 28.75 |
| 2048 | StreamingLLM | 22.54 | 22.46 | 35.35 | 33.36 | 23.11 | 13.30 | 26.60 | 20.67 | 26.47 | 66.00 | 85.92 | 42.00 | 2.81 | 26.58 | 55.92 | 51.58 | 34.67 | 18.75 |
| | SnapKv | 26.10 | 32.14 | 49.31 | 41.71 | 27.60 | 18.83 | 28.90 | **24.53** | 26.65 | 70.50 | **86.28** | 42.97 | 2.78 | **86.27** | 54.50 | 50.87 | 41.87 | 51.87 |
| | Radar | 26.31 | 31.89 | 49.55 | **43.18** | 26.83 | 17.47 | 30.38 | 23.30 | **27.18** | **71.50** | 86.15 | 42.49 | **3.39** | 72.25 | 55.76 | 51.87 | 41.22 | 60.00 |
| 4096 | StreamingLLM | 24.48 | 29.86 | 41.04 | 37.19 | 24.47 | 14.94 | 30.38 | 21.62 | 26.96 | 69.50 | 86.15 | 42.70 | 2.58 | 39.53 | **56.49** | 52.64 | 37.53 | 34.38 |
| | SnapKV | **26.31** | 33.61 | **50.35** | 42.67 | 27.89 | 18.74 | 30.73 | 24.24 | 27.00 | 71.00 | 86.25 | 43.14 | 2.60 | 86.10 | 54.19 | 51.22 | 42.25 | **61.25** |
| | Radar | 26.31 | **33.77** | 49.17 | 42.85 | **28.68** | 17.91 | **32.15** | 24.17 | 27.11 | 71.00 | 86.23 | **43.41** | 2.87 | 81.75 | 56.42 | **52.92** | **42.30** | 69.38 |

the Landmark attention (Mohtashami & Jaggi, 2023) and SubGen (Zandieh et al., 2024). Each experiment is conducted on one A100 GPU.

**Models.** We test a wide range of models in this experiment. We first choose the `Llama-7b` model (Touvron et al., 2023a) because Landmark attention is fine-tuned based on it. We also have `Llama-2-7b-chat-hf`, which is included in the original LongBench paper (Bai et al., 2024). Finally, we also test with `Mistral-7B-Instruct-v0.2` following the previous work (Li et al., 2024).

**Results.** The results are shown in Table 1. In all three subtables, we see that the vanilla attention mechanism generally achieves the highest scores on all tasks, given that it has all the information presented in the context. StreamingLLM is oftentimes the worst method because it only uses the recent $32+n_c$ tokens.

For the `Llama-7b` model (in Table 1a), we see that the Landmark attention method performs better than the vanilla method on QA tasks. This is likely due to the additional data in fine-tuning. However, the performance on other tasks is significantly deteriorated. For $H_2O$ and SnapKV, we observe similar results as in Li et al. (2024), where SnapKV is better than $H_2O$ in both average score and percentile. Among all the baselines, Radar achieves the highest average score and percentile ranking.

The results on `Llama-2-7b-chat-hf` are shown in Table 1b. Aligned with the observation in previous work (Li et al., 2024), SnapKV generally outperforms $H_2O$ in this setting. SubGen, albeit additionally saving memory, underperforms other methods on nearly all tasks. This indicates that its emphasis on memory saving and acceleration might be at the expense of generation quality. Our Radar achieves the highest average scores and average percentiles across all $n_c$ settings. Notably, Radar with 1024 middle tokens outperforms SnapKV with 2048 middle tokens in terms of average score. The results strongly suggest the effectiveness of our method in utilizing the context.

Table 1c shows the results for `Mistral-7B-Instruct-v0.2`.[2] Our method is consistently better than SnapKV when the ground-truth answers require long generation (e.g., the government report task). This not only suggests the effectiveness of Radar but also shows that Radar is utilizing different tokens at different steps of token generation. Remarkably, Radar achieves an average score close to the vanilla method with only 13% of the tokens used ($n_c = 4096$).

## 3.3 IN-DEPTH ANALYSES

**Generating without prompts.** In Section 3.1, we prefill 16,384 tokens for all runs to simulate the usage of prompts in real-world applications. However, some previous work are focused on non-conditional generation, where no prompt is provided. To compare the performance in this setting, we conduct experiments similar to Section 3.1 without the prompts. The results are shown in Figure 3. Note that we cannot compare with SnapKV because it is only applied to prompts.

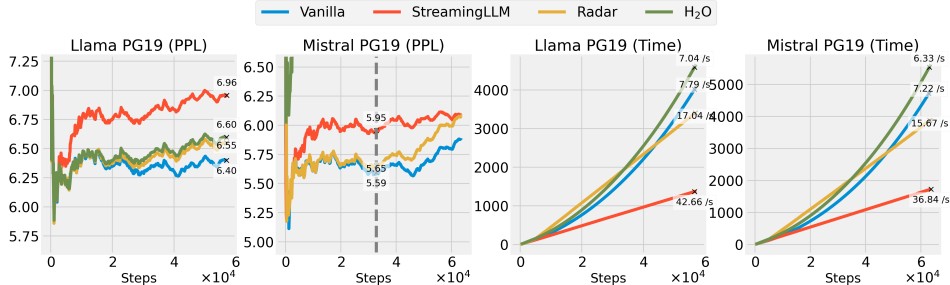

Figure 3: Generation without prompts.

As observed, $H_2O$ demonstrates a promising result on the Llama model—the overall perplexity is only worse than Radar by 0.05. However, the perplexity of the Mistral model starts to drastically deteriorate after hundreds of tokens. On the contrary, Radar continues to perform steadily. This experiment suggests the versatility of Radar, offering acceleration with or without prompts on a wide range of model architectures.

**The effect of $n$ and $k$.** Our method introduces hyper-parameters $n$ (the projection dimension for the random matrix) and $k$ (the number of top segments selected). We show the effect of these two hyper-parameters with the Figure 4a and Figure 4b, respectively.

Aligned with our theoretical prediction with Theorem 2, increasing $n$ improves the performance by providing better attention approximation. Similarly, increasing $k$ also improves the generation

---

[2]We skip the $H_2O$ method in this setting because it runs out of 40GB memory with prompts of 31.5K tokens.

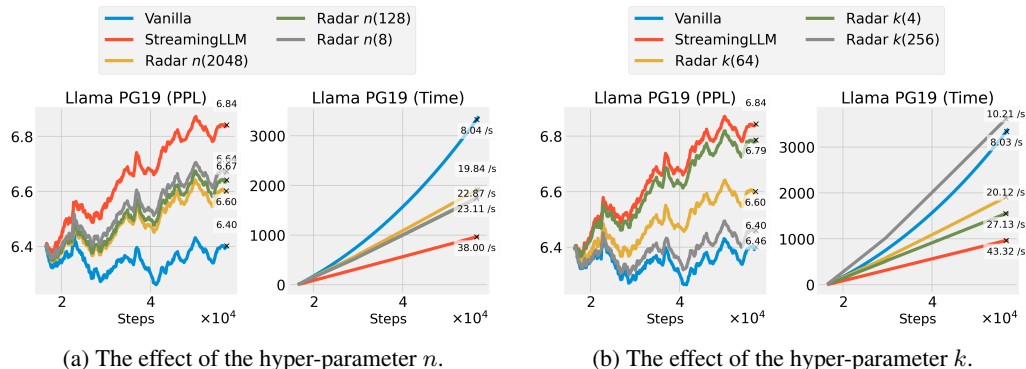

(a) The effect of the hyper-parameter $n$.      (b) The effect of the hyper-parameter $k$.

Figure 4: The effect of the hyper-parameters $n$ (the projection dimension for the random matrix) and $k$ (the number of top segments selected) introduced by Radar.

quality by using more tokens. However, a high $n$ or $k$ imposes the need for more memory and increases the computational overhead. Given these considerations, we use $n = 2048$ and $k = 64$ by default to balance the generation quality and hardware efficiency.

**Ablation study.** Our algorithm involves an approximated top-$k$ selection. To verify that such an approximation is functioning as intended, we conduct three ablation studies in Figure 5 by replacing it with three other strategies. The experiments are conduced with the Llama model on the PG-19 sample.

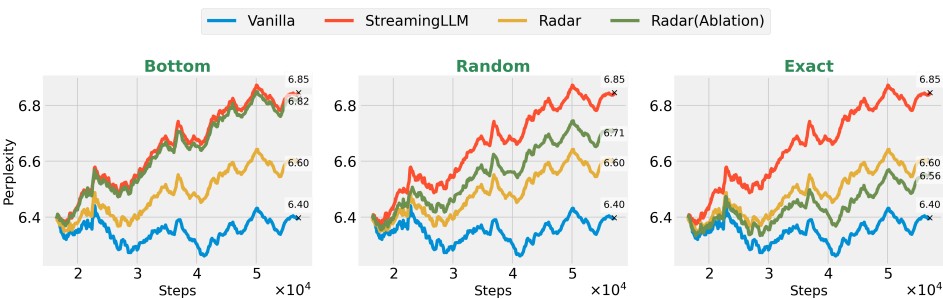

Figure 5: Ablation studies. Here, we compare with three different segment selection strategies.

As shown, when we select segments with the lowest approximated segment attention scores (left), the performance becomes similar to StreamingLLM, suggesting that the selected tokens with low segment attention scores are indeed not informative. In addition, our approximation is better than the random selection strategy (middle), showing that such an approximation is choosing more informative tokens than the "uneducated guess". Lastly, we see that our approximation has the closest performance compared with the exact segment search (right), indicating our method, albeit potentially missing some tokens, is a reasonable approximation given its low time complexity.

## 4 RELATED WORK

**Context reduction.** Previous literature shows that Transformers with sparse attention patterns can generate with fewer context tokens while not sacrificing the generation quality (Child et al., 2019; Beltagy et al., 2020; Zaheer et al., 2020; Roy et al., 2020; Chen et al., 2021). In addition to the success of sparse attention architectures, researchers also empirically demonstrate that the vanilla Transformers do not need all tokens in the context to correctly generate the current token (O'Connor & Andreas, 2021; Sun et al., 2021; Liu et al., 2024). Our work entails this discovery.

Based on the discovery, there are some closely related methods proposed Han et al. (2023). For instance, StreamingLLM (Xiao et al., 2024) uses only the local sliding window along with a few

beginning tokens to generate the current token. $H_2O$ (Zhang et al., 2023) proposes to additionally retain middle tokens based on a scoring function to maximize hit rate. SnapKV (Li et al., 2024) proposes to retain middle tokens based on the attention pooling in a parallel manner. All these methods, albeit reducing the context length, cannot recover the lost information once the context token is evicted. Moreover, they still suffer from the quadratic time complexity. By contrast, our method is able to dynamically retrieve any tokens in the context with a low time complexity. Recently, Zandieh et al. (2024) propose to compress the KV cache for memory and time efficiency, posing an opportunity for better algorithm designs.

**Efficient attention.** The quadratic computing time of the dot-product attention becomes a bottleneck in long-context generation. Searching for an efficient design of the attention mechanisms is a long-standing topic for Transformers (Wang et al., 2020; Kitaev et al., 2020). Recently, Landmark Attention (Mohtashami & Jaggi, 2023) proposes to use grouped softmax with a special token to reduce computation. Another line of work reduces the time complexity is through attention kernelization, which reformulates the dot-product attention mechanism as a kernel (Tsai et al., 2019; Katharopoulos et al., 2020; Peng et al., 2021; Zandieh et al., 2023). These methods can achieve linear complexity in both training and inference by recurrence. However, the parameters are required to be trained/fine-tuned to fit the new architectures. On the contrary, our method is training-free and can be applied to any pre-trained Transformers.

Instead of modifying the vanilla attention, recent literature also focuses on improving the efficiency in implementation. For example, Flash Attention series (Dao et al., 2022; Dao, 2023) leverage the tiling technique to avoid memory-bound attention operations. Memory-efficient attention (Rabe & Staats, 2021) reorders the attention calculation to allocate only the constant space for any context length. Our work is orthogonal to these methods and can be combined together.

**Long-context ability via retrieval.** In certain cases, Transformers can gain the long-context ability by retrieving related documents like retrieval-augmented generation (Lewis et al., 2020) based on $k$NN search. For instance, Memorizing Transformer (Wu et al., 2022) recurrently caches previous tokens in a $k$NN index. Unlimiformer (Bertsch et al., 2023) uses a $k$NN index to retrieve the encoded chunks in the encoder-decoder attention. However, these methods can only be used with the specialized architectures after training. They may also suffer from the exponential search quality degradation when the dimension increases (Beyer et al., 1999). Unlike these methods, Radar is a training-free method with a rigorous performance guarantee independent of the dimension.

## 5  CONCLUSION

In this work, we present Radar, a general and effective approach to accelerate inference for Transformers. Unlike previous methods that train other variants of the attention mechanism for replacement, Radar can accelerate any pre-trained Transformers without the need for training. In addition, we provide theoretical justifications that show Radar can reliably identify the most important tokens with high probability. More importantly, our empirical results across a wide range of tasks and models confirm the effectiveness of our method. Compared with prior training-free methods that evict tokens based on heuristics, Radar utilizes context tokens dynamically in a principled manner, resulting in significantly higher generation quality while achieving a lower time complexity.

We hope that Radar opens up opportunities for pre-trained Transformers to be applied out-of-the-box, without requiring expensive re-training, to long context settings (e.g. legal documents or financial transactions). We also hope our method further inspires advancements in high-quality generation with low time complexities for LLMs.

## ETHICS STATEMENT

Our research focuses on accelerating pre-trained Transformers to reduce power and time consumption in inference. As far as we are aware, it does not pose any new ethical or societal risks that require specific attention.

## REPRODUCIBILITY STATEMENT

In our paper, all of our models and datasets are publicly accessible. For the models, we download them from HuggingFace's hub using the following identifiers:

- `meta-llama/Meta-Llama-3.1-8B`
- `meta-llama/Llama-2-7b-chat-hf`
- `huggyllama/Llama-7b`
- `mistralai/Mistral-7B-v0.3`
- `mistralai/Mistral-7B-Instruct-v0.2`

For the datasets, we follow the previous work and similarly obtain them from HuggingFace:

- `THUDM/LongBench`
- `emozilla/pg19-test`
- `bigcode/the-stack-smol`

We run all of our experiments on A100 GPUs.

The procedure of Radar is explained in Algorithm 1 with the theoretical justification in Theorem 2.

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

## A  COMPLETE ALGORITHM

We provide the complete algorithm as follows.

---

**Algorithm 1** The overall algorithm of Radar

---

**Require:** head dimension $d \in \mathbb{N}$, projection dimension $n \in \mathbb{N}$, number of segment selection $k \in \mathbb{N}$
    $\triangleright$ Initialize the Radar state
1: Current context length $t \leftarrow 0$
2: Current segment range $c \leftarrow 0$
3: Unregistered buffer $\mathbb{W} \leftarrow \emptyset$
4: Sample $\boldsymbol{\Omega} = (\boldsymbol{\omega}_1, \ldots, \boldsymbol{\omega}_n)$ from $\mathcal{N}_{\boldsymbol{0,1}}$
5: Obtain the projection function $\phi_{\boldsymbol{\Omega}}$ according to Equation (4)
    $\triangleright$ Decoding procedure
6: **while** taking a new token with $\boldsymbol{q}, \boldsymbol{k}$ and $\boldsymbol{v}$ **do**
7:    $t \leftarrow t + 1$
8:    **if** $\sqrt{t} \in \mathbb{N}$ **then**
9:        $\triangleright$ Restructuring process descried in Section 2.2.
10:       $c \leftarrow \sqrt{t}$
11:       Pre-calculate $\bar{\phi}(\boldsymbol{k}_{l:l+c})$ for $l \in [1, 1+c, 1+2c, \ldots, 1+(c-1)c]$         $\triangleright O(t)$
12:       $\mathbb{W} \leftarrow \emptyset$
13:    **else**
14:       $\mathbb{W} \leftarrow \mathbb{W} \bigcup \{t\}$         $\triangleright$ Append to the buffer otherwise
15:    **end if**
       $\triangleright$ Querying procedure in $O(\sqrt{t})$
16:    **for** $l \in [1, 1+c, 1+2c, \ldots, 1+(c-1)c]$ **do**
17:       $a_{l:l+c} \leftarrow \phi_{\boldsymbol{\Omega}}^{\top}(\boldsymbol{q}) \bar{\phi}_{\boldsymbol{\Omega}}(\boldsymbol{k}_{l:l+c})$    $\triangleright$ Calculate segment attentions according to Equation (6)
18:    **end for**
19:    Pick indices $\mathbb{S}$ in the $k$ segments with the highest $a_{l:l+c}$         $\triangleright O(\sqrt{t})$
20:    $\mathbb{S} \leftarrow \mathbb{S} \bigcup \mathbb{W}$         $\triangleright$ Use the buffer as the sliding window
21:    **yield** $\mathrm{softmax}(\boldsymbol{q}^{\top} \boldsymbol{k}_{\mathbb{S}} / \sqrt{d}) \boldsymbol{v}_{\mathbb{S}}$         $\triangleright O(k\sqrt{t})$
22: **end while**

---

In the reconstruction step (lines 9-12), the time complexity is $O(t)$ because we have $c$ segments with each containing $c$ tokens. Since each segment representation $\bar{\phi}$ takes $O(c)$ to construct, line 11 takes at most $\sum_{i=1}^{c} O(c) = O(c^2)$ time. This is equivalent to $O(t)$ because $c = \sqrt{t}$ in line 10. As we will restructure the hierarchical approximation by at most $O(\sqrt{t})$ times (given by the condition in line 8), the overall time complexity of all restructuring steps (considering the outer loop containing lines 8-11) is $O(t^{3/2})$. When amortized to all $t$ steps, each step will take $O(\sqrt{t})$ on average.

For the query step (lines 16-18), it is straightforward to see the per-step time complexity is $O(\sqrt{t})$ because each time step computes $O(\sqrt{t})$ segment attention scores.

The final attention step (line 21) takes $O(k\sqrt{t})$ time because there are $k$ segments selected, each with $O(\sqrt{t})$ tokens.

Overall, we can see the per-step time complexity of Radar grows in $O(\sqrt{t})$.

## B  ESSENTIAL LEMMAS

**Lemma 3** (adapted from (Fan et al., 2015)). *Let* $X_1, \ldots, X_n$ *be i.i.d. random variables such that* $\mathbb{E}[X_i] = 0$, $X_i \leq b$. *For every* $\varepsilon \geq 0$, *we have*

$$\Pr\left(\frac{1}{n}\sum_{i=1}^{n} X_i \geq \varepsilon\right) \leq \exp\left(-\frac{n\varepsilon^2}{2c^2}\right) \tag{8}$$

$$\text{and } \Pr\left(-\frac{1}{n}\sum_{i=1}^{n} X_i \geq \varepsilon\right) \leq \exp\left(-\frac{n\varepsilon^2}{2c^2}\right). \tag{9}$$

*Here,*

$$c^2 := \begin{cases} \mathbb{E}[X_i^2], & \text{if } \mathbb{E}[X_i^2] \geq b^2, \\ \frac{1}{4}\left(b + \frac{\mathbb{E}[X_i^2]}{b}\right)^2, & \text{otherwise.} \end{cases} \tag{10}$$

*Proof.* Please refer to Corollary 2.7 in (Fan et al., 2015). $\qquad\square$

**Lemma 4.** *Let $Y_1, \ldots, Y_n$ be i.i.d. random variables such that $Y_i \geq 0$, $\mathbb{E}[Y_i] = \mu > 0$, and $\mathrm{Var}[Y_i] = \sigma^2$. We have*

$$\Pr\left(\left(\mu - \frac{1}{n}\sum_{i=1}^{n} Y_i\right) \geq \varepsilon\right) \leq \exp\left(-\frac{n\varepsilon^2}{2\max(\mu^2, \sigma^2)}\right) \tag{11}$$

$$\text{and } \Pr\left(\left(\frac{1}{n}\sum_{i=1}^{n} Y_i - \mu\right) \geq \varepsilon\right) \leq \exp\left(-\frac{n\varepsilon^2}{2\max(\mu^2, \sigma^2)}\right) \tag{12}$$

*for every $\varepsilon \geq 0$.*

*Proof.* Let $X_i := \mu - Y_i$. It is easy to verify that

1. $\mathbb{E}[X_i] = \mathbb{E}[\mu - Y_i] = \mu - \mathbb{E}[Y_i] = 0$, and

2. $X_i = \mu - Y_i \leq \mu$, and

3. $\mathbb{E}[X_i^2] = \mathbb{E}[Y_i^2] - \mu^2 = \mathrm{Var}[Y_i] = \sigma^2$.

We can then apply Lemma 3 on $X_1, \ldots, X_n$ with $b = \mu > 0$, which gives us

$$c^2 := \begin{cases} \mathbb{E}[X_i^2] = \sigma^2, & \text{if } \sigma^2 \geq \mu^2 > 0 \\ \frac{1}{4}\left(\mu + \frac{\mathbb{E}[X_i^2]}{\mu}\right)^2 \leq \mu^2, & \text{otherwise,} \end{cases} \tag{13}$$

summarized as $0 < c^2 \leq \max(\sigma^2, \mu^2)$. We can then obtain the proof by noticing that $\exp\left(-(n\varepsilon^2)/c^2\right)$ is monotonically increasing with $c^2 > 0$. $\qquad\square$

**Definition 5.** For any $\boldsymbol{u} \in \mathbb{R}^d$, we define a transformation function $f_{\boldsymbol{u}} : \mathbb{R}^d \to \mathbb{R}$ as $f_{\boldsymbol{u}}(\boldsymbol{\omega}) := \exp\left(\boldsymbol{\omega}^\top \boldsymbol{u} - \|\boldsymbol{u}\|^2/2\right)$.

**Lemma 6.** *Given any $\boldsymbol{u}, \boldsymbol{v}, \boldsymbol{z} \in \mathbb{R}^d$ and $\boldsymbol{\omega}$ as a random variable following $\boldsymbol{\omega} \sim \mathcal{N}_{\boldsymbol{0}, \boldsymbol{I}} \in \mathbb{R}^d$, we have the following properties for $f$:*

1. $f_{\boldsymbol{u}}(\boldsymbol{\omega}) \geq 0$, and

2. $\mathbb{E}[f_{\boldsymbol{u}}(\boldsymbol{\omega})] = 1$, and

3. $\mathbb{E}[f_{\boldsymbol{u}}(\boldsymbol{\omega}) f_{\boldsymbol{v}}(\boldsymbol{\omega})] = \exp\left(\boldsymbol{u}^\top \boldsymbol{v}\right)$, and

4. $\mathrm{Var}[f_{\boldsymbol{u}}(\boldsymbol{\omega}) f_{\boldsymbol{v}}(\boldsymbol{\omega})] = \exp\left(2\boldsymbol{u}^\top \boldsymbol{v}\right)\left(\exp\left(\|\boldsymbol{u} + \boldsymbol{v}\|^2\right) - 1\right)$.

5. $\mathrm{Cov}\left(f_{\boldsymbol{u}}(\boldsymbol{\omega}) f_{\boldsymbol{z}}(\boldsymbol{\omega}), f_{\boldsymbol{v}}(\boldsymbol{\omega}) f_{\boldsymbol{z}}(\boldsymbol{\omega})\right) = \exp\left((\boldsymbol{u} + \boldsymbol{v})^\top \boldsymbol{z}\right)\left(\exp\left((\boldsymbol{u} + \boldsymbol{z})^\top (\boldsymbol{v} + \boldsymbol{z})\right) - 1\right)$.

*Proof.* We prove each property as follows:

1. It is obvious that $\exp(x) \geq 0$ for all $x \in \mathbb{R}$.

2.

$$\mathbb{E}[f_{\boldsymbol{u}}(\boldsymbol{\omega})] := \mathbb{E}\left[\exp\left(\boldsymbol{\omega}^\top \boldsymbol{u} - \|\boldsymbol{u}\|^2/2\right)\right] \tag{14}$$

$$= \int \Pr(\boldsymbol{\omega}) \exp\left(\boldsymbol{\omega}^\top \boldsymbol{u} - \|\boldsymbol{u}\|^2/2\right) \mathrm{d}\boldsymbol{\omega} \tag{15}$$

$$=(2\pi)^{-d/2} \int \exp\left(-\|\boldsymbol{\omega}\|^2/2\right) \exp\left(\boldsymbol{\omega}^\top \boldsymbol{u} - \|\boldsymbol{u}\|^2/2\right) \mathrm{d}\boldsymbol{\omega} \tag{16}$$

$$=(2\pi)^{-d/2} \int \exp\left(\boldsymbol{\omega}^\top \boldsymbol{u} - \|\boldsymbol{u}\|^2/2 - \|\boldsymbol{\omega}\|^2/2\right) \mathrm{d}\boldsymbol{\omega} \tag{17}$$

$$=(2\pi)^{-d/2} \int \exp\left(-\|\boldsymbol{\omega} - \boldsymbol{u}\|^2/2\right) \mathrm{d}\boldsymbol{\omega} \tag{18}$$

$$= \Pr\left(\mathbb{R}^d\right) \tag{19}$$

$$=1. \tag{20}$$

3. This is first shown in Choromanski et al. (2021). We present it here for completeness.

$$\mathbb{E}[f_{\boldsymbol{u}}(\boldsymbol{\omega})f_{\boldsymbol{v}}(\boldsymbol{\omega})] = \mathbb{E}\left[\exp\left(\boldsymbol{\omega}^\top \boldsymbol{u} - \|\boldsymbol{u}\|^2/2\right)\exp\left(\boldsymbol{\omega}^\top \boldsymbol{v} - \|\boldsymbol{v}\|^2/2\right)\right] \tag{21}$$

$$= \mathbb{E}\left[\exp\left(\boldsymbol{\omega}^\top(\boldsymbol{u}+\boldsymbol{v}) - \|\boldsymbol{u}+\boldsymbol{v}\|^2/2\right)\exp\left(\boldsymbol{u}^\top \boldsymbol{v}\right)\right] \tag{22}$$

$$= \mathbb{E}[f_{\boldsymbol{u}+\boldsymbol{v}}(\boldsymbol{\omega})]\exp\left(\boldsymbol{u}^\top \boldsymbol{v}\right) \tag{23}$$

$$= \exp\left(\boldsymbol{u}^\top \boldsymbol{v}\right), \tag{24}$$

where the last line is from the second property. Notice that $\phi_{\boldsymbol{\Omega}}$ in the main text scales both $\boldsymbol{u}$ and $\boldsymbol{v}$ by $1/\sqrt[4]{d}$ before projection, resulting in $\mathbb{E}_{\boldsymbol{\Omega}}[\phi_{\boldsymbol{\Omega}}(\boldsymbol{u})^\top \phi_{\boldsymbol{\Omega}}(\boldsymbol{v})] = \exp\left(\boldsymbol{u}^\top \boldsymbol{v}/\sqrt{d}\right)$.

4. We first show that

$$\mathbb{E}[(f_{\boldsymbol{u}}(\boldsymbol{\omega})f_{\boldsymbol{v}}(\boldsymbol{\omega}))^2] \tag{25}$$

$$= \mathbb{E}\left[\exp\left(2\boldsymbol{\omega}^\top(\boldsymbol{u}+\boldsymbol{v}) - \|\boldsymbol{u}+\boldsymbol{v}\|^2\right)\exp\left(2\boldsymbol{u}^\top \boldsymbol{v}\right)\right] \tag{26}$$

$$= \mathbb{E}\left[\exp\left(2\boldsymbol{\omega}^\top(\boldsymbol{u}+\boldsymbol{v}) - \|2\boldsymbol{u}+2\boldsymbol{v}\|^2/2\right)\exp\left(2\boldsymbol{u}^\top \boldsymbol{v} + \|\boldsymbol{u}+\boldsymbol{v}\|^2\right)\right] \tag{27}$$

$$= \mathbb{E}[f_{2\boldsymbol{u}+2\boldsymbol{u}}(\boldsymbol{\omega})]\exp\left(2\boldsymbol{u}^\top \boldsymbol{v} + \|\boldsymbol{u}+\boldsymbol{v}\|^2\right) \tag{28}$$

$$= \exp\left(2\boldsymbol{u}^\top \boldsymbol{v} + \|\boldsymbol{u}+\boldsymbol{v}\|^2\right), \tag{29}$$

where the last line is from the second property. It is then clear that

$$\mathrm{Var}[f_{\boldsymbol{u}}(\boldsymbol{\omega})f_{\boldsymbol{v}}(\boldsymbol{\omega})] = \mathbb{E}[(f_{\boldsymbol{u}}(\boldsymbol{\omega})f_{\boldsymbol{v}}(\boldsymbol{\omega}))^2] - \left(\mathbb{E}[f_{\boldsymbol{u}}(\boldsymbol{\omega})f_{\boldsymbol{v}}(\boldsymbol{\omega})]\right)^2 \tag{30}$$

$$= \exp\left(2\boldsymbol{u}^\top \boldsymbol{v} + \|\boldsymbol{u}+\boldsymbol{v}\|^2\right) - \exp\left(2\boldsymbol{u}^\top \boldsymbol{v}\right) \tag{31}$$

$$= \exp\left(2\boldsymbol{u}^\top \boldsymbol{v}\right)\left(\exp\left(\|\boldsymbol{u}+\boldsymbol{v}\|^2\right) - 1\right). \tag{32}$$

5. More generally,

$$\mathrm{Cov}\left(f_{\boldsymbol{u}}(\boldsymbol{\omega})f_{\boldsymbol{z}}(\boldsymbol{\omega}), f_{\boldsymbol{v}}(\boldsymbol{\omega})f_{\boldsymbol{z}}(\boldsymbol{\omega})\right) \tag{33}$$

$$= \mathbb{E}[f_{\boldsymbol{u}}(\boldsymbol{\omega})f_{\boldsymbol{z}}(\boldsymbol{\omega})f_{\boldsymbol{v}}(\boldsymbol{\omega})f_{\boldsymbol{z}}(\boldsymbol{\omega})] - \mathbb{E}[f_{\boldsymbol{u}}(\boldsymbol{\omega})f_{\boldsymbol{z}}(\boldsymbol{\omega})]\mathbb{E}[f_{\boldsymbol{v}}(\boldsymbol{\omega})f_{\boldsymbol{z}}(\boldsymbol{\omega})]. \tag{34}$$

The first term is

$$\mathbb{E}[f_{\boldsymbol{u}}(\boldsymbol{\omega})f_{\boldsymbol{z}}(\boldsymbol{\omega})f_{\boldsymbol{v}}(\boldsymbol{\omega})f_{\boldsymbol{z}}(\boldsymbol{\omega})] \tag{35}$$

$$= \mathbb{E}\left[\exp\left(\boldsymbol{\omega}^\top(\boldsymbol{u}+\boldsymbol{z}) - \|\boldsymbol{u}+\boldsymbol{z}\|^2/2\right)\exp\left(\boldsymbol{u}^\top \boldsymbol{z}\right)\right. \tag{36}$$

$$\left.\exp\left(\boldsymbol{\omega}^\top(\boldsymbol{v}+\boldsymbol{z}) - \|\boldsymbol{v}+\boldsymbol{z}\|^2/2\right)\exp\left(\boldsymbol{v}\top \boldsymbol{z}\right)\right] \tag{37}$$

$$= \mathbb{E}[f_{\boldsymbol{u}+\boldsymbol{z}}(\boldsymbol{\omega})f_{\boldsymbol{v}+\boldsymbol{z}}(\boldsymbol{\omega})]\exp\left(\boldsymbol{u}^\top \boldsymbol{z} + \boldsymbol{v}^\top \boldsymbol{z}\right) \tag{38}$$

$$= \exp\left((\boldsymbol{u}+\boldsymbol{z})^\top(\boldsymbol{v}+\boldsymbol{z})\right)\exp\left(\boldsymbol{u}^\top \boldsymbol{z} + \boldsymbol{v}^\top \boldsymbol{z}\right). \tag{39}$$

The second term is

$$\mathbb{E}[f_{\boldsymbol{u}}(\boldsymbol{\omega})f_{\boldsymbol{z}}(\boldsymbol{\omega})]\mathbb{E}[f_{\boldsymbol{v}}(\boldsymbol{\omega})f_{\boldsymbol{z}}(\boldsymbol{\omega})] = \exp\left(\boldsymbol{u}^\top \boldsymbol{z} + \boldsymbol{v}^\top \boldsymbol{z}\right). \tag{40}$$

Overall,

$$\mathrm{Cov}\left(f_{\boldsymbol{u}}(\boldsymbol{\omega})f_{\boldsymbol{z}}(\boldsymbol{\omega}), f_{\boldsymbol{v}}(\boldsymbol{\omega})f_{\boldsymbol{z}}(\boldsymbol{\omega})\right) = \left(\exp\left((\boldsymbol{u}+\boldsymbol{z})^\top(\boldsymbol{v}+\boldsymbol{z})\right) - 1\right)\exp\left((\boldsymbol{u}+\boldsymbol{v})^\top \boldsymbol{z}\right). \tag{41}$$

We can thus conclude our proofs. $\qquad\qquad\qquad\qquad\qquad\qquad\qquad\qquad\qquad\qquad\qquad\quad\square$

**Definition 7** (Random feature approximation). Since Lemma 6 shows that $\mathbb{E}_{\boldsymbol{\omega}\sim\mathcal{N}_{\boldsymbol{0},\boldsymbol{I}}}[f_{\boldsymbol{u}}(\boldsymbol{\omega})f_{\boldsymbol{v}}(\boldsymbol{\omega})]$ $= \exp(\boldsymbol{u}^\top\boldsymbol{v})$, we can approximate $\exp(\boldsymbol{u}^\top\boldsymbol{v})$ by sampling a random variable $S_{\boldsymbol{u},\boldsymbol{v}}^{(n)}$, defined as

$$S_{\boldsymbol{u},\boldsymbol{v}}^{(n)} := \frac{1}{n}\sum_{i=1}^{n} f_{\boldsymbol{u}}(\boldsymbol{\omega}_i)f_{\boldsymbol{v}}(\boldsymbol{\omega}_i) \tag{42}$$

for $\boldsymbol{\omega}_i \sim \mathcal{N}_{\boldsymbol{0},\boldsymbol{I}}$. This random variable corresponds to the dot-product between two features using Equation (4).

**Lemma 8.** *Given any $\boldsymbol{u}_i, \boldsymbol{v}_i, \boldsymbol{z} \in \mathbb{R}^d$ for $i \in [1, c]$. Without loss of generality, we assume $\sum_{i=1}^{c}\exp(\boldsymbol{u}_i^\top\boldsymbol{z}) \le \sum_{i=1}^{c}\exp(\boldsymbol{v}_i^\top\boldsymbol{z})$. Let $\varepsilon := \frac{1}{c}\sum_{i=1}^{c}\left(\exp(\boldsymbol{v}_i^\top\boldsymbol{z}) - \exp(\boldsymbol{u}_i^\top\boldsymbol{z})\right)$ and $\zeta := \max\{\|\boldsymbol{u}_i\|, \|\boldsymbol{v}_i\|, \|\boldsymbol{z}\|\}$. We have $\sum_{i=1}^{c} S_{\boldsymbol{u}_i,\boldsymbol{z}}^{(n)} \le \sum_{i=1}^{c} S_{\boldsymbol{v}_i,\boldsymbol{z}}^{(n)}$ with a high confidence.*

*Proof.* If there is a constant $\alpha \in [0, 1]$ such that we have $0 \le \frac{1}{c}\sum_{i=1}^{c}\exp(\boldsymbol{v}_i^\top\boldsymbol{z}) - \frac{1}{c}\sum_{i=1}^{c}S_{\boldsymbol{v}_i,\boldsymbol{z}}^{(n)} \le \alpha\varepsilon$ and $0 \le \frac{1}{c}\sum_{i=1}^{c}S_{\boldsymbol{u}_i,\boldsymbol{z}}^{(n)} - \frac{1}{c}\sum_{i=1}^{c}\exp(\boldsymbol{u}_i^\top\boldsymbol{z}) \le (1-\alpha)\varepsilon$ at the same time, we can show

$$\sum_{i=1}^{c} S_{\boldsymbol{u},\boldsymbol{z}}^{(n)} \le \sum_{i=1}^{c} S_{\boldsymbol{v},\boldsymbol{z}}^{(n)} \iff \sum_{i=1}^{c}\left(\exp(\boldsymbol{v}^\top\boldsymbol{z}) - S_{\boldsymbol{v},\boldsymbol{z}}^{(n)} + S_{\boldsymbol{u},\boldsymbol{z}}^{(n)} - \exp(\boldsymbol{u}^\top\boldsymbol{z})\right) \le c\varepsilon. \tag{43}$$

Conversely, if $\frac{1}{c}\sum_{i=1}^{c}S_{\boldsymbol{u}_i,\boldsymbol{z}}^{(n)} > \frac{1}{c}\sum_{i=1}^{c}S_{\boldsymbol{v}_i,\boldsymbol{z}}^{(n)}$, we have either $\sum_{i=1}^{c}\exp(\boldsymbol{v}_i^\top\boldsymbol{z}) - \sum_{i=1}^{c}S_{\boldsymbol{v}_i,\boldsymbol{z}}^{(n)} > \alpha c\varepsilon$ or $\sum_{i=1}^{c}S_{\boldsymbol{u}_i,\boldsymbol{z}}^{(n)} - \sum_{i=1}^{c}\exp(\boldsymbol{u}_i^\top\boldsymbol{z}) > (1-\alpha)c\varepsilon$ for the $\alpha$. For the first case, we have

$$\Pr\left(\sum_{i=1}^{c}\exp(\boldsymbol{v}_i^\top\boldsymbol{z}) - \sum_{i=1}^{c}S_{\boldsymbol{v}_i,\boldsymbol{z}}^{(n)} \ge \alpha c\varepsilon\right) \tag{44}$$

$$\le \exp\left(-\frac{n\alpha^2\varepsilon^2 c^2}{2\max\left((\sum_{i=1}^{c}\mu_i)^2, \sum_{i=1,j=1}^{i=c,j=c}\mathrm{Cov}(S_{\boldsymbol{v}_i,\boldsymbol{z}}^{(n)}, S_{\boldsymbol{v}_j,\boldsymbol{z}}^{(n)})\right)}\right) \tag{45}$$

$$\le \exp\left(-\frac{n\alpha^2\varepsilon^2 c^2}{2\sum_{i=1,j=1}^{i=c,j=c}\mathrm{Cov}(S_{\boldsymbol{v}_i,\boldsymbol{z}}^{(n)}, S_{\boldsymbol{v}_j,\boldsymbol{z}}^{(n)}) + \mu_i\mu_j}\right) \tag{46}$$

$$= \exp\left(-\frac{n\alpha^2\varepsilon^2 c^2}{2\sum_{i=1,j=1}^{i=c,j=c}\exp((\boldsymbol{v}_i+\boldsymbol{z})^\top(\boldsymbol{v}_j+\boldsymbol{z}))\exp(\boldsymbol{v}_i^\top\boldsymbol{z}+\boldsymbol{v}_j^\top\boldsymbol{z})}\right) \tag{47}$$

$$\le \exp\left(-\frac{n\alpha^2 c^2\varepsilon^2}{2(\sum_{i=1}^{i=c}\exp(\boldsymbol{v}_i\boldsymbol{z}))^2\exp(2\zeta^2)}\right) \tag{48}$$

$$=: \exp\left(-\frac{n\alpha^2\varepsilon^2 c^2}{2\beta_{\boldsymbol{v}}^2\exp(2\zeta^2)}\right). \tag{49}$$

Here, we define $\beta_{\boldsymbol{v}} := \sum_{i=1}^{c}\exp(\boldsymbol{v}_i\boldsymbol{z})$ for simplicity. Similarly, we have for the second case:

$$\Pr\left(\sum_{i=1}^{c}S_{\boldsymbol{u}_i,\boldsymbol{z}}^{(n)} - \sum_{i=1}^{c}\exp(\boldsymbol{u}_i^\top\boldsymbol{z}) \ge (1-\alpha)c\varepsilon\right) \tag{50}$$

$$\le \exp\left(-\frac{n(1-\alpha)^2 c^2\varepsilon^2}{2\max\left((\sum_{i=1}^{c}\mu_i)^2, \sum_{i=1,j=1}^{i=c,j=c}\mathrm{Cov}(S_{\boldsymbol{u}_i,\boldsymbol{z}}^{(n)}, S_{\boldsymbol{u}_j,\boldsymbol{z}}^{(n)})\right)}\right) \tag{51}$$

$$\le \exp\left(-\frac{n(1-\alpha)^2 c^2\varepsilon^2}{2\sum_{i=1,j=1}^{i=c,j=c}\exp((\boldsymbol{u}_i+\boldsymbol{z})^\top(\boldsymbol{u}_j+\boldsymbol{z}))\exp(\boldsymbol{u}_i^\top\boldsymbol{z}+\boldsymbol{u}_j^\top\boldsymbol{z})}\right) \tag{52}$$

$$\le \exp\left(-\frac{n(1-\alpha)^2 c^2\varepsilon^2}{2(\sum_{i=1}^{i=c}\exp(\boldsymbol{u}_i\boldsymbol{z}))^2\exp(2\zeta^2)}\right) \tag{53}$$

$$=: \exp\left(-\frac{n(1-\alpha)^2 c^2 \varepsilon^2}{2\beta_{\boldsymbol{u}}^2 \exp(2\zeta^2)}\right). \tag{54}$$

We choose $\alpha := \beta_{\boldsymbol{v}}/(\beta_{\boldsymbol{v}} + \beta_{\boldsymbol{u}})$, which means

$$\exp\left(-\frac{n\alpha^2 \varepsilon^2 c^2}{2\exp(2\zeta^2)\beta_{\boldsymbol{v}}^2}\right) = \exp\left(-\frac{nc^2\varepsilon^2}{2\exp(2\zeta^2)(\beta_{\boldsymbol{v}} + \beta_{\boldsymbol{u}})^2}\right) \tag{55}$$

and

$$\exp\left(-\frac{n(1-\alpha)^2 c^2 \varepsilon^2}{2\exp(2\zeta^2)\beta_{\boldsymbol{u}}^2}\right) = \exp\left(-\frac{nc^2\varepsilon^2}{2\exp(2\zeta^2)(\beta_{\boldsymbol{v}} + \beta_{\boldsymbol{u}})^2}\right). \tag{56}$$

Putting all together, we have

$$\Pr\left(\sum_{i=1}^{c} S_{\boldsymbol{u}_i, \boldsymbol{z}}^{(n)} \geq \sum_{i=1}^{c} S_{\boldsymbol{v}_i, \boldsymbol{z}}^{(n)}\right) \tag{57}$$

$$\leq \Pr\left(\sum_{i=1}^{c} \left(\exp(\boldsymbol{v}_i^\top \boldsymbol{z}) - S_{\boldsymbol{v}_i, \boldsymbol{z}}^{(n)}\right) \geq \alpha c\varepsilon\right) + \Pr\left(\sum_{i=1}^{c} \left(S_{\boldsymbol{u}_i, \boldsymbol{z}}^{(n)} - \exp(\boldsymbol{u}_i^\top \boldsymbol{z})\right) \geq (1-\alpha)c\varepsilon\right) \tag{58}$$

$$\leq 2\exp\left(-\frac{nc^2\varepsilon^2}{2\exp(2\zeta^2)(\beta_{\boldsymbol{v}} + \beta_{\boldsymbol{u}})^2}\right) \tag{59}$$

$$=: 2\exp\left(-\left(\frac{\sqrt{n}(a_{\boldsymbol{v}}' - a_{\boldsymbol{u}}')}{\sqrt{2}\exp(\zeta^2)}\right)^2\right), \tag{60}$$

where we use $a_{\boldsymbol{v}}' := \beta_{\boldsymbol{v}}/(\beta_{\boldsymbol{v}} + \beta_{\boldsymbol{u}})$ and $a_{\boldsymbol{u}}' := \beta_{\boldsymbol{u}}/(\beta_{\boldsymbol{v}} + \beta_{\boldsymbol{u}})$ for simplicity. $\qquad\square$

**Lemma 9.** *Given any $\boldsymbol{u}_i, \boldsymbol{v}_i, \boldsymbol{z} \in \mathbb{R}^d$ for $i \in [1, c]$. Without loss of generality, we assume $\sum_{i=1}^{c} \exp\left(\boldsymbol{u}_i^\top \boldsymbol{z}/\sqrt{d}\right) \leq \sum_{i=1}^{c} \exp\left(\boldsymbol{v}_i^\top \boldsymbol{z}/\sqrt{d}\right)$. Let $\varepsilon := \frac{1}{c}\sum_{i=1}^{c} \left(\exp\left(\boldsymbol{v}_i^\top \boldsymbol{z}/\sqrt{d}\right) - \exp\left(\boldsymbol{u}_i^\top \boldsymbol{z}/\sqrt{d}\right)\right)$ and $\zeta := \max\{\|\boldsymbol{u}_i\|, \|\boldsymbol{v}_i\|, \|\boldsymbol{z}\|\}$. We have $\sum_{i=1}^{c} S_{\boldsymbol{u}_i, \boldsymbol{z}}^{(n)} \leq \sum_{i=1}^{c} S_{\boldsymbol{v}_i, \boldsymbol{z}}^{(n)}$ with confidence at least*

$$1 - 2\exp\left(-\left(\frac{\sqrt{n}(a_{\boldsymbol{v}}' - a_{\boldsymbol{u}}')}{\sqrt{2}\exp\left(\zeta^2/\sqrt{d}\right)}\right)^2\right). \tag{61}$$

*Proof.* This is a direct application of Lemma 8 by scaling $\boldsymbol{v}_i, \boldsymbol{u}_i$, and $\boldsymbol{z}$ in Lemma 8 by $1/\sqrt[4]{d}$. $\quad\square$

## C  THE PROOF OF THEOREM 2

Now we provide the proof for Theorem 2 in the main text.

**Theorem 2.** *Let Radar be at a state of $c$ segments where each segment contains $c$ tokens. Without loss of generality, assume the first segment $\boldsymbol{k}_{1:1+c}$ has the highest attention score $a_{1:1+c}$ w.r.t. the current query $\boldsymbol{q}$. With confidence at least $1 - \delta$, we have*

$$\phi_{\boldsymbol{\Omega}}^\top(\boldsymbol{q})\bar{\phi}_{\boldsymbol{\Omega}}(\boldsymbol{k}_{1:c+1}) = \max_l \phi_{\boldsymbol{\Omega}}^\top(\boldsymbol{q})\bar{\phi}_{\boldsymbol{\Omega}}(\boldsymbol{k}_{l:c+l}) \tag{7}$$

*if the minimum gap $a_{1:1+c} - \max_l a_{l:l+c} \geq \frac{1}{c}\exp\left(\zeta^2/\sqrt{d}\right)\sqrt{\frac{8\log(2(c-1)/\delta)}{n}}$. Here, $\zeta$ is the maximum norm among $\boldsymbol{k}_i$ and $\boldsymbol{q}$, whereas $l \in [1, 1+c, 1+2c, \ldots, 1+(c-1)c]$ is the starting index of each segment.*

*Proof.* Let $\beta_i$ denote the summation of the unormalized attention weights in $i$th segment $\sum_{j=1}^{c} \exp\left(\boldsymbol{q}^\top \boldsymbol{k}_{ci+j}/\sqrt{d}\right)$. We know that $\beta_1 \geq \beta_i$ for all $i > 1$ by our assumption. Let $\varepsilon_i := (\beta_1 - \beta_i)/c$ be the difference between the first and the $i$th segment. According to Lemma 9, the probability of confusion between segment 1 and $i$ has

$$\delta_i \leq 2\exp\left(-\left(\frac{\sqrt{n}(a_1' - a_i')}{\sqrt{2}\exp\left(\zeta^2/\sqrt{d}\right)}\right)^2\right), \tag{62}$$

where $a_i' := \beta_i/(\beta_1 + \beta_i)$. The real attention score for $i$th segment is $a_i = \beta_i / \sum_{j=1}^{c} \beta_j$, meaning

$$\delta_i \leq 2\exp\left(-\left(\frac{\sqrt{n}\varepsilon_i}{\sqrt{2}\exp\left(\zeta^2/\sqrt{d}\right)(a_1 + a_i)}\right)^2\right) \leq 2\exp\left(-\left(\frac{\sqrt{n}\varepsilon}{\sqrt{2}\exp\left(\zeta^2/\sqrt{d}\right)(2a_1)}\right)^2\right), \tag{63}$$

where $\varepsilon := \min_i \varepsilon_i$ is the minimum gap.

The probability of making at least one confusion is

$$\delta \leq \sum_{i=2}^{c} \delta_i \leq 2(c-1)\exp\left(-\left(\frac{\sqrt{n}\varepsilon}{\sqrt{2}\exp\left(\zeta^2/\sqrt{d}\right)(2a_1)}\right)^2\right). \tag{64}$$

This is equivalent to

$$\varepsilon \leq a_1 \exp\left(\zeta^2/\sqrt{d}\right)\sqrt{\frac{8\log(2(c-1)/\delta)}{n}}, \tag{65}$$

which concludes the proof by taking the contraposition and noticing that $a_1 \geq 1/c$. $\square$

## D  ADDITIONAL EXPERIMENTS

In Section 3.1, we mainly test our methods against StreamingLLM (Xiao et al., 2024) and the vanilla attention (Vaswani et al., 2017). In fact, we also consider $H_2O$ (Zhang et al., 2023) and SnapKV (Li et al., 2024) as our baselines. Following the same settings as in Section 3.1, we show their performance in Figure 6.

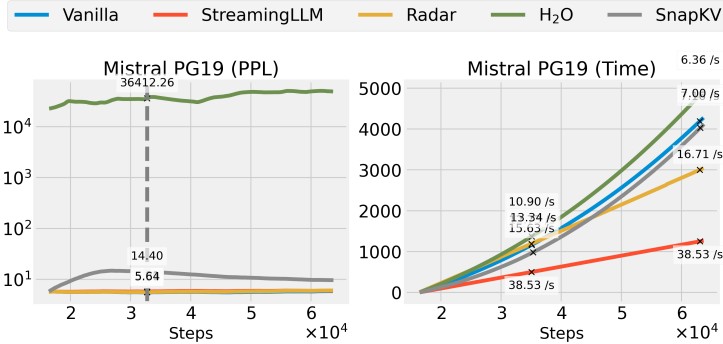

Figure 6: Failures of $H_2O$ and SnapKV on Mistral.

As observed, both methods suffer from the considerable degeneration under this setting.[3] The performance of $H_2O$, similar to its non-conditional generation with Mistral models (in Section 3.3), has

---

[3]It should be pointed out that we do not consider their failures are due to our replication, because we successfully reproduce their results under the settings in their papers (e.g., Section 3.2 and Section 3.3).

a extremely high perplexity, indicating its unsuitability beyond a limited model architectures like Llama. SnapKV, albeit having a low complexity at the beginning, shows a considerable loss degradation during generation. This aligns with our observations in Section 3.2 where SnapKV generally underperforms Radar when the ground-truth generation is long. The results also suggest that Radar generalizes better than other baselines. We hypothesize that this is because methods like H2O and SnapKV heuristically use the accumulated attention scores as the indicator of the importance. The heuristics may lead to evicting the current unimportant tokens which will be very useful later for new tokens, especially when each head is responsible for multiple query spaces (e.g., grouped-query attention (Ainslie et al., 2023) used in Llama 3 and Mistral models). Evidently, we observed that both methods tend to perform worse on these models. By contrast, Radar is based on a proven approximation of the vanilla attention. The error bound is well-established in our Theorem 2 regardless of the model architectures and weights, making it more versatile in applications.

# E   VISUALIZING THE APPROXIMATION

In Theorem 2, we theoretically show the correctness of our method. To provide an intuition for the approximation, we visualized the Radar approximation using Llama 3 on the 100 tokens (after 1 sink token) of PG-19 partitioned in 10 segments. The results are displayed in Figure 7.

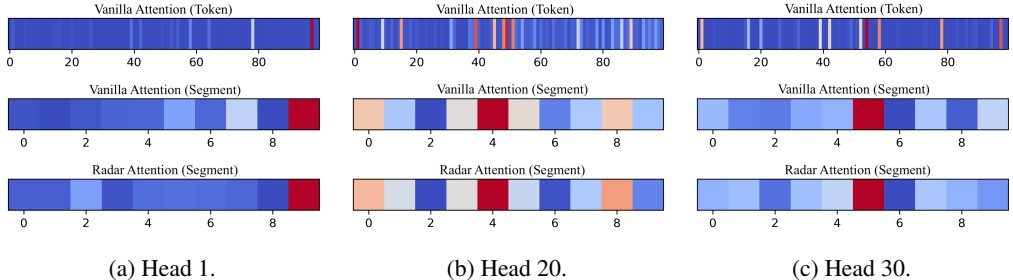

(a) Head 1.  (b) Head 20.  (c) Head 30.

Figure 7: Visualization of the vanilla token-level attention (top), the corresponding segment attention (middle), and the approximated segment attention by Radar (bottom). Here we use three heads in the first layer as an example. The dark blue and red represent low and high attention scores, respectively.

It is straightforward to see that Radar (the bottom row) provides a reasonable approximation of the real segment attention (the middle row). Notably, Radar not only captures the top segments but also correctly flags the runner-up segments, suggesting its ability in identifying multiple important segments simultaneously. Empirically, we found that 34.38% of Radar's approximation successfully flags the top segment among the 10 segments with $k = 1$. When we increase $k = 3$, the successful rate is 62.5%. These are considerable higher than other segmental selection strategies based on heuristics. For example, by selecting the most recent segments (i.e., extending the sliding window for StreamingLLM (Xiao et al., 2024)), the corresponding rates are 18.75% and 46.88%. The random selection strategy is expected to achieve 10% and 30%. Both methods underperform Radar significantly.

# F   DISCUSSION ON FUTURE WORK

**Non-contiguous segmenting strategies.**   In this work, we propose to use contiguous tokens to construct a segment. This is generally a safe practice because the important tokens are sparse in a sequence (as visualized in Figure 7 and previous work like StreamingLLM Xiao et al. (2024)). In addition, by selecting the surrounding tokens around the important ones, we could provide the context for the model to correctly identify the semantics for these tokens. Our design could naturally achieve this goal, while other methods like SnapKV (Li et al., 2024) need to explicitly apply

smoothing techniques to encourage the surrounding selection. Nevertheless, we consider that there could exist better segmenting mechanisms for Radar by developing an adaptive method. We hope this paper serves as the foundation for the future exploration of this direction.

**Memory-efficient Radar.** Our method mainly focuses on accelerating the decoding efficiency of modern Transformer models. However, Radar does not save runtime memory. In fact, it uses marginally more memory (scaled at a $O(\sqrt{t})$ rate) to maintain the segment representations. Therefore, we consider the memory-efficient version of Radar as a promising future direction. For example, one could exploit the hierarchical structure of Radar to align with the memory hierarchy by off-loading. By doing this, we hope Radar could match the effective memory saving with previous memory-efficient decoding algorithms (Zhang et al., 2023; Li et al., 2024; Zandieh et al., 2024).

**Training with Radar.** Our method is designed as a training-free acceleration algorithm to reduce the training cost for broader users. However, we expect training with this kind of hierarchical approximation could yield better results. We leave this to future exploration.

