# OpenReview forum: "Radar: Fast Long-Context Decoding for Any Transformer"
_ICLR.cc/2025/Conference — ICLR 2025 Poster_

### Official Review · Reviewer_LfUC · 2024-10-28

**Soundness:** 3
**Presentation:** 3
**Contribution:** 3
**Rating:** 8
**Confidence:** 3

**Summary:**

This paper proposes an algorithm for accelerating LLM inference by efficiently selecting of important tokens. Its headline claim is a reduction from $\mathcal{O}(t^2)$ sequence length complexity to $\mathcal{O}(t^{3/2})$.

**Strengths:**

I am not an expert in efficient approximations to softmax attention, so I cannot confidently speak to the novelty of this algorithm relative to other methods. I will confine my comments to the results as presented. The proposed algorithm is simple, and the experiments demonstrate that it achieves a seemingly reasonable balance of accuracy and efficiency. Moreover, I found the paper to be reasonably clearly written.

**Weaknesses:**

In my mind, the paper would be stronger if there was a clearer link between Theorem 2 and the experiments. Namely, unless I've missed something, the authors don't actually measure whether their algorithm retrieves the correct top-k index set in their experiments. It would be useful to do so, and to comment on whether by those results the bound in Theorem 2 is practically useful or not. It would also be useful to know something about the distribution of attention scores, which would affect what value of $k$ is reasonable. More precise (and in some sense more mechanistic) quantifications could help the reader get a better sense of why your method works.

**Questions:**

- Do you have any intuition for why the H20 and SnapKV methods perform so poorly when applied to the Mistral PG19 model (Appendix C)? The difference is quite striking, so it would be useful to have at least a hypothesis for why these methods do not generalize across architectures. This could also provide useful context for why the proposed method successfully generalizes.

- There appears to be a typo on Line 837, as the square brackets enclose the equality.

- It might be useful to state Lemma 1 in terms of concentration to the expectation at large $n$, as that could give a bound on error in a practical setting.

- Given that the claimed time-complexity improvement is in the exponent of a power law, it would be useful to use logarithmic scales for time in the figures, so that one can see how tight the bound is.

- It could be semantically useful to use a different color scheme for Figure 4, to distinguish the fact that here different dimensions are being used rather than different algorithms.

---

> ### Author Response · Authors · 2024-11-23
>
> > In my mind, the paper would be stronger if there was a clearer link between Theorem 2 and the experiments. Namely, unless I've missed something, the authors don't actually measure whether their algorithm retrieves the correct top-k index set in their experiments. It would be useful to do so, and to comment on whether by those results the bound in Theorem 2 is practically useful or not. It would also be useful to know something about the distribution of attention scores, which would affect what value of $k$ is reasonable. More precise (and in some sense more mechanistic) quantifications could help the reader get a better sense of why your method works.
>
> We appreciate the reviewer's insightful question. Theorem 2 is a worst-case analysis that provides a theoretical framework for understanding dot product approximation via random projection. The bound offers insights into factors impacting attention approximation, which could guide future work in this area to derive tighter bounds for specific practical settings. Empirically, we found that the approximation works well in many practical cases (34% hit@1 and 63% hit@3), which performed better than StreamingLLM (19% / 47%) and random (10% / 30%) on the first 100 tokens (after the first one) of the PG-19 dataset using the Llama 3.1 model. Visualizations of how the approximation works also confirm this. We have added this new analysis to  Appendix D.
>
> > Do you have any intuition for why the H20 and SnapKV methods perform so poorly when applied to the Mistral PG19 model (Appendix C)? The difference is quite striking, so it would be useful to have at least a hypothesis for why these methods do not generalize across architectures. This could also provide useful context for why the proposed method successfully generalizes.
>
> Thanks for asking this. We do have a hypothesis of why Radar generalizes better than H2O and SnapKV. Specifically, both H2O and SnapKV heuristically use the accumulated attention scores as the indicator of the importance. However, the heuristics may lead to evicting the current unimportant tokens which will be very useful later for new tokens, especially when each head is responsible for multiple query spaces (i.e., in group-query attention used in Llama 3 and Mistral models). Indeed, we observed that both methods tend to perform worse on these models. By contrast, our Radar is a proven approximation of the vanilla attention method. The error bound is well-established in our Theorem 2 regardless of the model architectures and weights, making it more versatile in applications. We have added this discussion in Appendix C.
>
>
> > There appears to be a typo on Line 837, as the square brackets enclose the equality.
>
> Yes, you are correct. Thank you for identifying this typo. We have fixed it in the PDF.
>
> > It might be useful to state Lemma 1 in terms of concentration to the expectation at large
> $n$, as that could give a bound on error in a practical setting.
>
> Yes, we agree that stating it in terms of concentration will be more informative. However, since building the concentration inequality for a single approximation does not directly justify our method, we only stated its expectation here to make it easier to understand. Nevertheless, we expect that one can easily build such an inequality following the same procedure as our Lemma 8 based on the provided properties in our Lemma 6.
>
> > Given that the claimed time-complexity improvement is in the exponent of a power law, it would be useful to use logarithmic scales for time in the figures, so that one can see how tight the bound is.
> > It could be semantically useful to use a different color scheme for Figure 4, to distinguish the fact that here different dimensions are being used rather than different algorithms.
>
>
> We appreciate these detailed suggestions on writing. We have tried both the log-scale time ([this anonymous link](https://pasteboard.co/NJXpKnCjefmJ.png)) and other color schemes ([this anonymous link](https://pasteboard.co/4Ct2jXllYkYF.png)). Since there are some overlapping curves in both figures that might hinder readability, we temporarily maintained our original figures. However, we are grateful for this suggestion and will explore other designs in the next revision.
>
> ---
>
> Overall, we would like to thank the reviewer for recognizing our contributions and outlining the highly detailed comments. Based on the suggestion, we have improved our paper by visualizing the theory in Theorem 2. We believe that our revised paper further resolves concerns. We are more than happy to discuss further if the reviewer has further suggestions.

---

> > ### Comment · Reviewer_LfUC · 2024-11-23
> >
> > Thank you for your detailed reply to my (sometimes naive) questions! I think you've adequately addressed my questions and the concerns of the other referees, so I'll raise my score to 8.

---

> > > ### Author Response · Authors · 2024-11-27
> > >
> > > We are grateful for your detailed questions and for recognizing the adequacy of our responses to your and other reviewers' concerns. We appreciate your decision to raise the scores and are pleased that our efforts have met your expectations. We remain open to any further feedback you may have.

---

### Official Review · Reviewer_VET1 · 2024-10-31

**Soundness:** 2
**Presentation:** 3
**Contribution:** 2
**Rating:** 3
**Confidence:** 4

**Summary:**

The paper proposes a new method to approximate the softmax attention mechanism in transfomer architecture. The method works by segmenting the key embeddings and then for each query picking a few segments which have highest attention scores on average and computing the attention scores for the keys in those segments exactly. The mehtod uses random features proposed in the Performer paper  to estimate the average attention score of each segment.

**Strengths:**

The paper proposes a practical and easily implementable algorithm that demonstrates strong performance compared to existing streaming attention and sliding window attention methods.

**Weaknesses:**

The paper lacks a comparison with several relevant works focused on attention acceleration. Notably, the following papers explore sparse, low-rank, and combinations of these approaches with near-linear runtimes for long context, and it remains unclear how the proposed method measures up against them:

- `Chen, Beidi, et al. "Scatterbrain: Unifying sparse and low-rank attention." Advances in Neural Information Processing Systems 34 (2021): 17413-17426.`
- `Zandieh, Amir, et al. "Kdeformer: Accelerating transformers via kernel density estimation." International Conference on Machine Learning. PMLR, 2023.`
- Han, Insu, et al. "Hyperattention: Long-context attention in near-linear time." arXiv preprint arXiv:2310.05869 (2023).
- `Zandieh, Amir, et al. "SubGen: Token Generation in Sublinear Time and Memory." arXiv preprint arXiv:2402.06082 (2024).`
- `Roy, Aurko, et al. "Efficient content-based sparse attention with routing transformers." Transactions of the Association for Computational Linguistics 9 (2021): 53-68.`

Additionally, this method does not tackle the memory concerns regarding storing key-value embeddings in the KV cache; it requires retaining all key and value embeddings in the cache during the generation phase.

**Questions:**

- How does your method compare against the aforementioned works in experimental evaluations?

- The paper does not clarify how tokens are segmented. Do you segment the stream of tokens into contiguous chunks, or do you employ a different ordering method?

Additionally, I would like to note: please try to address my concerns, and rest assured that I will consider raising my score if you provide sufficient experimental results during the rebuttal.

---

> ### Author Response · Authors · 2024-11-23
> **Official Comment by Authors (1/2)**
>
> > The paper lacks a comparison with several relevant works focused on attention acceleration. Notably, the following papers explore sparse, low-rank, and combinations of these approaches with near-linear runtimes for long context, and it remains unclear how the proposed method measures up against them. Additionally, this method does not tackle the memory concerns regarding storing key-value embeddings in the KV cache; it requires retaining all key and value embeddings in the cache during the generation phase. How does your method compare against the aforementioned works in experimental evaluations?
>
>
> Thank you for suggesting the papers. In the initial submission, we mainly followed the baselines and experimental settings in recent papers StreamingLLM (at ICLR 2024) and SnapKV (at NeurIPS 2024). Based on the comment, we now have included all of the suggested papers in the related work section. However, we would like to point out that many of them are not directly comparable as they use different settings. Specifically, our method is a post-training acceleration method without using any data. Among 5 suggested papers, there are 3 methods (Scatterbrain, KDEformer, and Routing Transformers) that require training. As discussed in lines 504-512, we cannot directly use these models as baselines because they have different architectures trained with their unique data. Our method is specifically proposed to tackle the inefficiency of the vanilla Transformer attention for inference after training.
>
> The remaining 2 methods (HyperAttention and the recent SubGen) are more related. However, their experiments were conducted on ChatGLM and LongChat, whereas our method and all our baselines use Llama and Mistral models. This creates difficulty in conducting fair comparisons given the limited amount of time during the discussion. Nevertheless, we are able to replicate SubGen (the newer method from the same group as HyperAttention) on the already included LongBench dataset using the Llama models. The preliminary results are as follows
>
>
> **Llama 2 with maximum 1024+32 tokens used in each step**
>
>
> |  | qasper (Single QA) | multifieldqa_en (Single QA) | hotpotqa (Multi QA) | 2wikimqa (MultiQA) | multi_news (Summarization) | trec (Few-shot) | passage_retrieval_en (Synthetic) | lcc (Code) |
> |---|---|---|---|---|---|---|---|---|
> | Vanilla | $\underline{17.78}$ | $\underline{36.13}$ | $\textbf{34.06}$ | $\underline{27.53}$ | $\textbf{25.99}$ | $\textbf{64.00}$ | $\textbf{12.00}$ | $\textbf{58.36}$ |
> | StreamingLLM | $15.13$ | $21.95$ | $28.67$ | $24.56$ | $24.57$ | $61.00$ | $4.50$ | $56.59$ |
> | SubGen | $13.70$ | $29.07$ | $19.1$ | $21.36$ | $21.99$ | $37.00$ | $5.08$ | $45.48$ |
> | Radar | $\textbf{19.32}$ | $\textbf{37.20}$ | $\underline{33.70}$ | $\textbf{27.60}$ | $\underline{25.66}$ | $\underline{63.50}$ | $\underline{10.00}$ | $\underline{57.70}$ |
>
> The results show that our method performs consistently better than the baseline (including SubGen). We believe that the results strongly suggest the capability of our method.
>
>
> However, we acknowledge that these results should not discard the values of previous work as their main target is memory efficiency, whereas Radar focuses on time efficiency and accuracy. We have added this discussion in Appendix E and will be committed to updating the discussion when we have complete results of the suggested baselines.

---

> > ### Author Response · Authors · 2024-11-23
> > **Official Comment by Authors (2/2)**
> >
> > > The paper does not clarify how tokens are segmented. Do you segment the stream of tokens into contiguous chunks, or do you employ a different ordering method?
> >
> >
> > Thank you for raising this question. We use contiguous chunks as implied by Equation 5 and Algorithm 1. Based on your question, we have added a discussion in Appendix E. Essentially, our reasoning for this choice is as follows. First, this is naturally a safe assumption because the important tokens are sparse in a sequence (as shown in our new Figures and previous work like StreamingLLM). In addition, by selecting the surrounding tokens around the important ones, we could provide the context for the model to correctly identify the semantics for these tokens. Our design could naturally achieve this goal, while other methods like SnapKV need to explicitly apply smoothing techniques to encourage the “surrounding selection”. Nevertheless, we consider that there could exist better segmenting mechanisms for Radar by developing an adaptive method. We hope this paper serves as the foundation and leaves the exploration of this direction to future work.
> >
> >
> >
> >
> > ---
> >
> >
> > We would like to thank the reviewer for providing the insightful comments and pointing to several related papers. The major concern of the reviewer is that the performance of other methods could be better. During the discussion, we have conducted some key experiments to resolve this concern. The results show that our method consistently outperforms other baselines. We have also added the discussion on the potential reasons in our paper. We believe that the quality of our paper is improved by a large margin based on the reviewer’s comments. We are also grateful to the reviewer for expressing the willingness to change scores. Should the reviewer have any further comments, please feel free to let us know.

---

> > > ### Author Response · Authors · 2024-11-30
> > >
> > > Dear reviewer. We haven’t received your further comments since our response one week ago. Given the deadline is approaching, we would like to kindly ask that if your concerns are sufficiently addressed. In any case, we are fully committed to answering all the questions you might have.

---

### Official Review · Reviewer_Un5J · 2024-11-02

**Soundness:** 3
**Presentation:** 2
**Contribution:** 3
**Rating:** 6
**Confidence:** 3

**Summary:**

This work introduces a novel training-free approach that accelerates inference by dynamically identifying the most important context tokens. The paper also provides proofs of algorithmic correctness and includes a time complexity analysis. The experimental results indicate competitive performance.

**Strengths:**

1. The problem addressed is significant and appealing, focusing on a crucial aspect of LLM performance.
2. The paper presents a novel approach for accelerating LLM inference by selecting the most important context tokens.
3. The authors provide detailed proofs of algorithmic correctness and analyze the time complexity.
4. The experimental results are competitive, highlighting the effectiveness of the proposed method.

**Weaknesses:**

1. The analysis of time complexity needs more detail, particularly regarding lines 8 to 11 of Algorithm 1. This dynamic restructuring step may take more time than $O(t)$. Based on my understanding, this section could have a total time complexity of $\sum_{c=1}^{\sqrt{t}} c^2 = O(t^{3/2})$. Thus, the time required for calculating the importance score of a single segment is not constant.

**Questions:**

See weakness. If I made mistakes, please tell me and I am glad to give a high score.

---

> ### Author Response · Authors · 2024-11-23
>
> > The analysis of time complexity needs more detail, particularly regarding lines 8 to 11 of Algorithm 1. This dynamic restructuring step may take more time than $O(t)$. Based on my understanding, this section could have a total time complexity of $\sum_{c=1}^{\sqrt{t}} c^2 = O(t^{3/2})$. Thus, the time required for calculating the importance score of a single segment is not constant.
>
> In line 11, the time complexity is $O(t)$ because we have $c$ segments with each containing $c$ tokens. Since each segment representation $\bar\phi$ takes $O(c)$ (according to Eqn. 5) to construct, line 11 takes at most $\sum_{i=1}^c O(c) = O(c^2)$ time. This is equivalent to $O(t)$ because $c=\sqrt{t}$ in line 10.
>
>
> The overall restructuring steps (considering the outer loop containing lines 8-11) is $O(t^{3/2})$ time, as we will restructure the hierarchical approximation by at most $O(\sqrt{t})$ times (because of the condition in line 8). When amortized to all $t$ steps, each step will take $O(\sqrt{t})$ on average, matching the querying complexity. We analyzed this in lines 205-207.
>
>
> We hope this could better illustrate the time complexity analyses.
>
> ---
>
> We sincerely appreciate your efforts in verifying our calculation and capturing the technical details. By double-checking the relevant parts, we are confident in our analyses of the time complexities. We hope our response answers your concerns. We are also happy to answer any further questions.

---

> ### Comment · Reviewer_Un5J · 2024-11-25
>
> Thank you for your reply. I understand your perspective that each segment representation takes $O(c)$. However, I believe Line 11 calculates $c$ segments for each $c$, with each segment containing $c$ tokens. Therefore, the computation cost for Line 11 at a specific $c$ is $O(c) \times c = O(c^2)$.
>
> Let’s break this down clearly:
>
> For a given $c$, there are $c$ segments to compute. Each segment contains $c$ tokens, requiring $O(c)$ to process. Hence, the total cost for that iteration is $c \times O(c) = O(c^2)$.
>
> When summing over all $c$ in the range $[1, \sqrt{t}]$, the total time complexity is:
> $
> \sum_{c=1}^{\sqrt{t}} c^2 = O(t^{3/2}),
> $
> as the sum of squares formula confirms.
>
> If you disagree with $O(c^2)$ for Line 11, could you clarify where you think this step is optimized or reduced? I’m open to revisiting the logic if there’s something I missed.

---

> > ### Author Response · Authors · 2024-11-25
> >
> > Thanks for showing your thought!
> >
> > We agree that line 11 takes $O(c^2)$ time and we stated "line 11 takes at most $O(c^2)$ time" in the response. But this doesn’t entail that "the time required for calculating the importance score of a single segment is not constant". The reason is Line 11 is the *construction* step, whereas the importance score calculation happens during *querying* (line 17). To summarize:
> >
> > * The overall construction time complexity is $O(t^{3/2})$, amortized to $O(\sqrt{t})$ per step.
> > * The query time complexity to calculate the importance score of one segment *is* constant (as shown in line 17).  Consequently, querying all $O(\sqrt{t})$ segments  results in $O(\sqrt{t})$ time per step.
> >
> > We appreciate your quick response and showing your thought clearly. We hope this addresses your question.

---

> ### Comment · Reviewer_Un5J · 2024-11-27
>
> Thank you for your response. That said, I believe the paper still faces some challenges in terms of clarity and organization, which may impact its overall readability and impact. For instance, the $O(t)$ complexity mentioned in line 11 is somewhat unclear and could benefit from further elaboration.
> Nevertheless, I acknowledge the significant effort put into addressing the concerns raised, and I have adjusted my score accordingly to reflect this progress.
> Lastly, I have one final question: how do we select $k$ indices into $S$ from $c$ segments? Are we randomly selecting one index from the top $k$ segments with the highest average attention scores? How to ensure that the selected index in the segment related to high attention score with query?

---

> > ### Comment · Reviewer_Un5J · 2024-11-27
> >
> > In addition, can the authors explain why we need training process in inference process when we use Kernelized Attention?

---

> > > ### Author Response · Authors · 2024-11-30
> > >
> > > > I believe the paper still faces some challenges in terms of clarity and organization, which may impact its overall readability and impact. For instance, the $O(t)$ complexity mentioned in line 11 is somewhat unclear and could benefit from further elaboration.
> > >
> > > Thank you for giving concrete suggestions on improving the presentation of our paper. Although the PDF is not editable at this moment, we are dedicated to incorporating your suggestion in the revision by further elaborating the analysis.
> > >
> > > > Lastly, I have one final question: how do we select $k$ indices into $S$ from $c$ segments? Are we randomly selecting one index from the top $k$ segments with the highest average attention scores? How to ensure that the selected index in the segment related to high attention score with query?
> > >
> > > Yes, we select the top-$k$ segments into $S$. While the selection utilizes the random projection technique, the selection is not random. Instead, the random projection and transformation approximates the real segment attention scores (shown in Theorem 2). Based on the theory, selecting according to the approximated scores would tend to recover the segments with the highest scores in the original attention. We hope this response addresses your question.
> > >
> > > > In addition, can the authors explain why we need training process in inference process when we use Kernelized Attention?
> > >
> > > In previous methods using kernelized attention, the approximation error makes it infeasible to directly perform inference. For example, the resulting approximation could have negative attention scores [1], significantly changing the softmax behavior. Therefore, it is usually required to train such models before deploying them for inference. By contrast, our method uses real attention scores after selection, reducing the approximation error.
> > >
> > > ---
> > >
> > > Thank you again for acknowledging our previous response. We are fully committed to answer additional questions should you have any.
> > >
> > > [1] Peng, Hao, et al. "Random Feature Attention." International Conference on Learning Representations. 2020.

---

### Official Review · Reviewer_v6nX · 2024-11-02

**Soundness:** 4
**Presentation:** 3
**Contribution:** 3
**Rating:** 8
**Confidence:** 3

**Summary:**

Transformer are the core of LLMs. This paper introduces a novel approach coined "Radar", which accelerates inference and is even training-free. Theoretical results are provided and the effectiveness is shown by numerical experiments.

**Strengths:**

* The new approach selects the most important tokens with high probability.
* It accelerates inferences for transformers.
* The method does not need to be trained.
* Theoretical results are provided.
* Also the numerical results are convincing.

**Weaknesses:**

no weaknesses

**Questions:**

no questions

---

> ### Author Response · Authors · 2024-11-23
>
> Thank you for your detailed and positive evaluation of our work. We greatly appreciate your recognition of the strengths of our approach! Encouraged by the positive comments, we have updated the PDF which further enhances the strength of this paper. Please feel free to let us know should you have any further questions.

---

> > ### Comment · Reviewer_v6nX · 2024-11-23
> > **Comment on Rebuttal**
> >
> > Dear authors, thank you for your comment. Since I already gave very good ratings, I will leave them as they are.

---

> > > ### Author Response · Authors · 2024-11-27
> > >
> > > We are grateful for your positive assessment and the high ratings you have given. Your support is appreciated, and we remain open to any further feedback you may wish to share.

---

### Official Review · Reviewer_RGXx · 2024-11-03

**Soundness:** 4
**Presentation:** 4
**Contribution:** 4
**Rating:** 8
**Confidence:** 4

**Summary:**

The paper describes a training-free approach to speed up attention computation over a long context, by essentially performing a hierarchical search for the 'closest' tokens in the attention dot-product. Tokens are grouped into segments, and attention is restricted to the closest segments --- which itself is found through dot product attention to a 'segment'-level averaged projected feature.

The paper provides theoretical analysis of this approximation, and experiments show speedups over vanilla attention without significant drop in quality.

**Strengths:**

- The paper is very well written and motivated.
- The hierarchical search approach is novel, as is the use of averaged random projection features to allow identifying segments without retraining.
- The theoretical analysis is a nice addition to the paper and is insightful in understanding why we expect the approach to work.
- The experimental results are solid, and the approach appears to achieve real speedups when run on GPUs (not just in theoretical flops).

**Weaknesses:**

- There seems to be an assumption that segments should be made of contiguous tokens. This might be a safe assumption, but should be discussed. I can imagine an 'adversarial' setting where the ideal attention matrix would have been centered on a subset of tokens spaced far apart (i.e., each in a different segment). The theoretical analysis could perhaps have focused on this as well (how likely it is for the top-k segments to include the top-k tokens).

- It would have been nice to include some examples of the true and approximated attention maps (as heatmap figures, and perhaps also as text examples). We only see performance numbers at the final task, but not directly how well attention is being approximated.

**Questions:**

In addition to speaking to the weaknesses listed above, it would be good if the authors could discuss whether they think this kind of hierarchical approximation to attention might work even better if it were included during training, not just as a post-training step.

---

> ### Author Response · Authors · 2024-11-23
>
> > There seems to be an assumption that segments should be made of contiguous tokens. This might be a safe assumption, but should be discussed. I can imagine an 'adversarial' setting where the ideal attention matrix would have been centered on a subset of tokens spaced far apart (i.e., each in a different segment). The theoretical analysis could perhaps have focused on this as well (how likely it is for the top-k segments to include the top-k tokens).
>
> Thank you for the insightful comment. We have added a discussion in Appendix E. Essentially, our reasoning for this choice is as follows. First, as you mentioned, this is naturally a safe assumption because the important tokens are sparse in a sequence (as shown in our new Figures and previous work like StreamingLLM). In addition, by selecting the surrounding tokens around the important ones, we could provide the context for the model to correctly identify the semantics for these tokens. Our design could naturally achieve this goal, while other methods like SnapKV need to explicitly apply smoothing techniques to encourage the “surrounding selection”. Nevertheless, we consider that there could exist better segmenting mechanisms for Radar by developing an adaptive method. We hope this paper serves as the foundation and leaves the exploration of this direction to future work.
>
> > It would have been nice to include some examples of the true and approximated attention maps (as heatmap figures, and perhaps also as text examples). We only see performance numbers at the final task, but not directly how well attention is being approximated.
>
> Thank you for the suggestion. We have added the headmap figures in Appendix D based on the suggestion. We believe this further enhances the strength of our paper.
>
> > In addition to speaking to the weaknesses listed above, it would be good if the authors could discuss whether they think this kind of hierarchical approximation to attention might work even better if it were included during training, not just as a post-training step.
>
> Thank you for the question. Due to the time constraint, we could not complete relevant experiments. However, we have discussed this in Appendix E, stating  “We expect training with this kind of hierarchical approximation could yield better results.” and “We leave this to future exploration.”
>
> ---
>
>
> Thank you for reviewing our paper. We highly appreciate the comments for acknowledging our contributions and novelty. We have also included the related discussions in our paper. We believe these discussions further strengthen our submission. Please do not hesitate to let us know for more questions.

---

> > ### Comment · Reviewer_RGXx · 2024-11-26
> >
> > Thanks for the responses. Having read the other reviews and author responses as well, I still have a positive view of the paper and happy to keep my rating.

---

> > > ### Author Response · Authors · 2024-11-27
> > >
> > > We appreciate your continued positive view of our paper and the time you have dedicated to reviewing our responses alongside those of other reviewers. Your thorough consideration is greatly valued, and we are pleased to have your support.

---

### Author Response · Authors · 2024-11-23

Dear all reviewers,

We sincerely thank you for your efforts and insightful comments. We have updated the PDF file to reflect our responses to the discussion. All our edits are marked in orange to distinguish them from the initial version. Specifically, we mainly made the following changes:

- Appendix C provides additional discussions on the performances of H2O and SnapKV.
- Appendix D provides the attention heatmap figures.
- Appendix E provides a discussion on potential directions for future work.

We hope our responses clear out the misunderstandings in the initial reviews. Please feel free to raise any additional questions.

Best regards, \
The Authors

---

### Meta-Review · Area_Chair_Yysm · 2024-12-15

**Metareview:**

The paper describes a training-free approach to speed up attention computation over a long context. The idea is to perform a hierarchical search for the closest tokens in the attention dot-product. Tokens are grouped into segments, and attention is restricted to the closest segments. The authors provide a theoretical analysis and rather extensive numerical experiments.

The reviewers praised the well-motivated problem, the novel hierarchical search approach, the insightful theoretical analysis and the convincing numerical experiments. Some issues were raised concerning the analysis of the time complexity and the comparison with related work. However, the rebuttal from the authors addressed such concerns in a sufficiently convincing way.

Hence, I recommend acceptance of the paper. I strongly encourage the authors to incorporate the latest parts of the discussion with Reviewer Un5J in the final version, as well as the complete results of the baselines suggested by Reviewer VET1.

**Additional Comments On Reviewer Discussion:**

Reviewer Un5J raised some concerns about the analysis of the time complexity. This led to a detailed discussion. At the end of this discussion, the reviewer still expressed some concerns in terms of the clarity and organization of the paper, but recognized the efforts of the authors. I see this mainly as a problem of readability that can be addressed by local edits in the final version.

Reviewer VET1 raised some concerns about the comparison with related work. In response, the authors provided additional numerical evidence confirming the advantage of the proposed method with respect to one of the additional baselines. Other baselines were either not directly comparable, or not compared due to the short rebuttal period. While the reviewer did not further engage in a discussion, I find the additional results compelling enough to justify an acceptance.

---

### Decision · Program_Chairs · 2025-01-22

Accept (Poster)